# Myelin dystrophy impairs signal transmission and working memory in a multiscale model of the aging prefrontal cortex

Sara Ibañez[1,2,3†], Nilapratim Sengupta[1,4†], Jennifer I Luebke[1], Klaus Wimmer[2,3‡], Christina M Weaver[4*‡]

[1]Department of Anatomy & Neurobiology, Boston University Chobanian & Avedisian School of Medicine, Boston, United States; [2]Centre de Recerca Matemàtica, Edifici C, Campus Bellaterra, Bellaterra, Spain; [3]Departament de Matemàtiques, Universitat Autònoma de Barcelona, Edifici C, Bellaterra, Spain; [4]Department of Mathematics, Franklin and Marshall College, Lancaster, United States

*For correspondence: christina.weaver@fandm.edu

†Co-first authors: All authors consent that for their CV/Resume and other purposes, co-first authors may list themselves in either order

‡Co-senior authors

Competing interest: The authors declare that no competing interests exist.

**Abstract** Normal aging leads to myelin alterations in the rhesus monkey dorsolateral prefrontal cortex (dlPFC), which are positively correlated with degree of cognitive impairment. It is hypothesized that remyelination with shorter and thinner myelin sheaths partially compensates for myelin degradation, but computational modeling has not yet explored these two phenomena together systematically. Here, we used a two-pronged modeling approach to determine how age-related myelin changes affect a core cognitive function: spatial working memory. First, we built a multi-compartment pyramidal neuron model fit to monkey dlPFC empirical data, with an axon including myelinated segments having paranodes, juxtaparanodes, internodes, and tight junctions. This model was used to quantify conduction velocity (CV) changes and action potential (AP) failures after demyelination and subsequent remyelination. Next, we incorporated the single neuron results into a spiking neural network model of working memory. While complete remyelination nearly recovered axonal transmission and network function to unperturbed levels, our models predict that biologically plausible levels of myelin dystrophy, if uncompensated by other factors, can account for substantial working memory impairment with aging. The present computational study unites empirical data from ultrastructure up to behavior during normal aging, and has broader implications for many demyelinating conditions, such as multiple sclerosis or schizophrenia.

## eLife assessment

This manuscript reports a **valuable** computational study of the effects of axon de-myelination and re-myelination on action potential speed and propagation failure. The manuscript presents **solid** evidence for the effects of de- and re-myelination in different models of working memory, with potential implications in disorders such as multiple sclerosis. The exposition of the manuscript is targeted for researchers interested in biophysical models of cognitive deficits.

## Introduction

Normal aging often leads to impairment in some cognitive domains, as evidenced by reduced performance on learning and memory tasks in both humans (**Albert, 1993**; **Salthouse et al., 2003**; **Fisk and Sharp, 2004**; **Rhodes, 2004**; **Sorel and Pennequin, 2008**) and non-human primates (**Moore et al.,**

**Figure 1.** Electron photomicrographs (transverse sections) depicting age-related alterations in myelinated nerve fibers of area 46 of the rhesus monkey dorsolateral prefrontal cortex (dlPFC). (**A**) Neuropil from a 10-year-old monkey. Healthy and compact myelin is visible as thick outlines surrounding nerve fibers which have been sectioned at their internodes. (**B**) Neuropil from a 27-year-old monkey. Arrows indicate dystrophic myelin surrounding nerve fibers, presenting a splitting of the major dense line of the myelin sheaths (left and right arrows) and balloons (left and middle arrows). Scale bar = 5 μm. Images are from the archives of Alan Peters and prepared as in *Peters and Sethares, 2002*.

*2006*; *Shamy et al., 2011*; *Moore et al., 2017*; *Comrie et al., 2018*; *Chang et al., 2022*; *Moore et al., 2023*). In the rhesus monkey, age-related working memory decline is accompanied by sublethal structural and functional changes in vascular elements, individual pyramidal neurons, glial cells, and white matter pathways (reviews: *Hof and Morrison, 2004*; *Luebke et al., 2010*; *Peters and Kemper, 2012*; *Morrison and Baxter, 2012*). It is well documented that cortical neurons do not die during normal aging but rather undergo a number of morphological and physiological alterations, particularly in the monkey dorsolateral prefrontal cortex (dlPFC), the critical cortical circuit for working memory. For example, during normal aging Layer 3 pyramidal neurons in rhesus monkey dlPFC exhibit a significant loss of dendritic spines and synapses (*Chang et al., 2005*; *Peters et al., 2008*; *Chang et al., 2022*), and electrophysiological changes observed both in vitro (*Chang et al., 2005*; *Ibañez et al., 2019*; *Chang et al., 2022*) and in vivo during a spatial working memory task (*Wang et al., 2011*). Perhaps most strikingly, extensive myelin dystrophy during normal aging has been observed in both gray and white matter (*Peters et al., 2001*; *Bowley et al., 2010*; review: *Peters, 2007*), including in monkey dlPFC (*Peters and Sethares, 2002*; *Peters and Sethares, 2003*; review: *Peters, 2009*). Ultrastructural studies reveal that 3–6% of myelin sheaths in dlPFC exhibit age-related alterations including splitting of the major dense line of the myelin sheath, balloons, and redundant myelin (*Figure 1*; *Peters and Sethares, 2002*). Remyelination has also been observed across the adult lifespan in monkey dlPFC and there is a 90% increase in the number of paranodal profiles in aged versus young monkeys, indicating higher numbers of internodal myelin sheaths with aging (*Peters and Sethares, 2003*). Aged subjects also had a significant proportion of abnormally short and thin myelin sheaths (*Peters and Sethares, 2003*). The hypothesized mechanism to explain these findings (review:*Peters, 2009*) is that myelin degradation begins as oligodendrocytes degenerate due to oxidative stress, and that axons accumulate dense inclusions in spaces between the lamellae of their associated myelin sheaths. As oligodendrocytes die, the associated sheaths detach from the axolemma, leaving bare axonal sections (complete demyelination). Subsequently, surviving mature oligodendrocytes remyelinate the bare segments, but with shorter and thinner sheaths. It is highly plausible that the altered sheaths lead to a slowdown of signal propagation that contributes to cognitive slowing/impairment with aging. Indeed, several of the changes that pyramidal neurons undergo with aging correlate with the degree of observed cognitive impairment (review: *Luebke et al., 2010*; *Peters and Kemper, 2012*; *Shobin et al., 2017*; *Moore et al., 2023*), including myelin dystrophies and remyelination (*Peters and Sethares, 2002*; *Peters and Sethares, 2003*; *Dimovasili et al., 2023*). However, which changes are the key determinants of age-related cognitive decline has not yet been firmly established (*Konar et al., 2016*; *Motley et al., 2018*; *Cleeland et al., 2019*). This is in part due to the difficulty of isolating

some individual neuronal features (e.g. firing rate, synapses, myelin) empirically, while controlling for concomitant changes in others. Thus, computational models become essential to predict how age-related changes in individual neurons affect cognitive impairment.

Numerous modeling studies have explored the potential relationships between axon parameters and action potential (AP) conduction velocity (CV) (*Rushton, 1951*; *Goldman and Albus, 1968*; *Brill et al., 1977*; *Moore et al., 1978*; *Waxman, 1980*; *Chomiak and Hu, 2009*). Others have sought the most appropriate way to model the axon (e.g. *Blight, 1985*; *Richardson et al., 2000*; *McIntyre et al., 2002*; *Gow and Devaux, 2008*; *Dekker et al., 2014*). Demyelination has been modeled frequently in the context of disease (review: *Coggan et al., 2015*), showing that loss of myelin leads to slower AP CV and sometimes to AP failure. Remyelination of axons with shorter, thinner internodes decreases CV in models of both large- and small-diameter axons (*Lasiene et al., 2008*; *Powers et al., 2012*; *Scurfield and Latimer, 2018*). However, to our knowledge, the effects of widespread demyelination and remyelination on AP propagation in a broad population of axons have not been explored systematically. There have also been studies of how CV and myelin alterations impact network synchrony in phase oscillator models (*Karimian et al., 2019*; *Pajevic et al., 2014*; *Noori et al., 2020*; *Pajevic et al., 2023*), but not in spiking neural networks that can predict the mechanisms underlying working memory. Our recent network model revealed that the empirically observed age-related increase in AP firing rates in prefrontal pyramidal neurons (modeled through an increased slope of the *f-I* curve) and loss of up to 30% of both excitatory and inhibitory synapses (modeled as a decrease in connectivity strength) can lead to working memory impairment (*Ibañez et al., 2019*), but this model did not incorporate the known changes to myelin structure that occur during normal aging.

This study unites models at two different scales – multicompartment models of dlPFC pyramidal neurons and a spiking network model simulating spatial working memory – to investigate how age-related myelin degradation (represented by demyelination) and remyelination affect signal transmission and working memory precision. Higher degrees of demyelination led to slower propagation and eventual failure of APs along the axons of the multicompartment models. In the network models, an increase in AP failure rate resulted in progressive working memory impairment, whereas slower conduction velocities, in the range observed in the multicompartment models, had a negligible effect. Sufficient remyelination of all previously demyelinated segments led to a recovery of signal transmission and working memory performance. However, our study indicates that empirically observed levels of myelin changes, if uncompensated by other factors, would lead to substantial working memory impairment with aging.

## Results

### Progressive demyelination causes CV slowing and AP failures in model neurons

To simulate myelin alterations in individual neurons, we adapted our multicompartment model tuned to data from rhesus monkey dlPFC (*Rumbell et al., 2016*) by attaching an axon model that captured nodes and detailed myelinated segments (*Gow and Devaux, 2008*; *Scurfield and Latimer, 2018*). Myelinated segments included an internode with adjacent juxtaparanodes and paranodes, and tight junctions between the innermost myelin lamella and axolemma (see Methods; *Figure 2A and B*). We applied demyelination and remyelination perturbations to a cohort of 50 young (control) neuron models, with axonal parameters varying within biologically plausible ranges (*Table 1*; *Figure 2—figure supplement 1A and B*). To simulate demyelination, we removed lamellae from selected myelinated segments; for remyelination we replaced a fraction of myelinated segments by two shorter and thinner segments with a node in between. As such, a 'fully remyelinated axon' had all the demyelinated segments subsequently remyelinated, but with fewer lamellae and additional nodes compared to the unperturbed control case, consistent with empirical observations (*Peters, 2009*). The CV in control models varied across the cohort and in response to myelin alterations (*Figure 2C*; *Figure 2—figure supplement 1*). Myelin alterations could also cause AP failures. CV changes and AP failures were more sensitive to variations along some dimensions of the parameter space than to others (e.g. myelinated segment length versus axon diameter), explored further below.

AP propagation was progressively impaired as demyelination increased (*Figure 3*): CV became slower, eventually leading to AP failure. Removing 25% of lamellae had a negligible effect on CV,

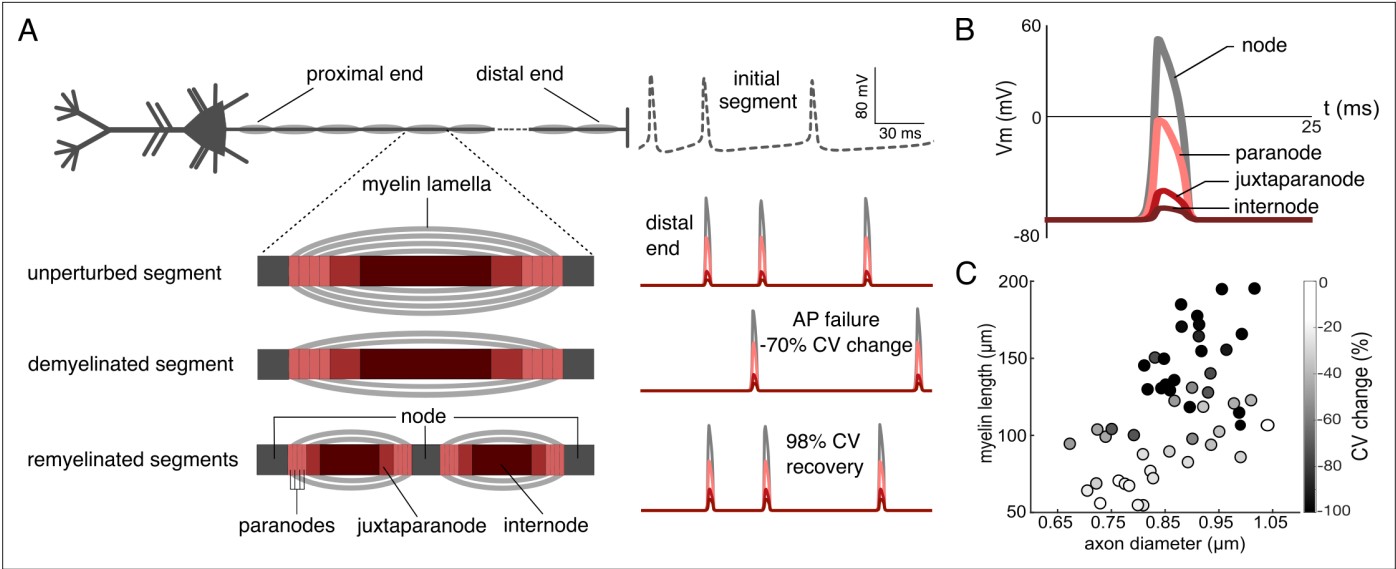

**Figure 2.** Action potential (AP) transmission in the single neuron model. (**A**) Cartoon of the model with a close-up view of unperturbed, demyelinated, and remyelinated segments (not to scale). The paranodes, juxtaparanodes, and internodes (shown in different shades of red) were insulated by myelin lamellae, adjacent to unmyelinated nodes (dark gray). During demyelination, lamellae were removed from a subset of segments; middle cartoon shows two lamellae remaining, indicating 50% lamellae removed relative to an unperturbed myelinated segment. During remyelination, select myelinated segments were replaced with two shorter myelinated segments separated by a new node; bottom cartoon shows remyelination with 50% of lamellae restored relative to unperturbed segments. At right are shown membrane potential traces simulated at the initial segment (top, dashed line) and near the distal end of one axon (here, 1.9 cm long) in the unperturbed, demyelinated, and remyelinated cases. Traces correspond to signals in a distal node and subsequent paranode, juxtaparanode, and internode respectively (colors indicating the axonal sections as in left panels). Demyelinating 75% of segments by removing 50% of their lamellae resulted in a 70% reduction in conduction velocity (CV), and failure of one AP. Remyelination of all affected segments with the same 50% of lamellae recovered the failed AP, and 98% of the CV delay relative to the demyelinated case (in 1 of the 30 simulated trials). (**B**) Close-up view of an AP simulated in the distal end of the unperturbed axon: suprathreshold in the node and subthreshold along the myelinated segment, indicating saltatory conduction. (**C**) Distribution of the 50 models of the cohort across two dimensions of parameter space: myelinated segment length and axon diameter. Grayscale shade of each model represents the mean CV change across three demyelination conditions: 25%, 50%, 75% of segments losing lamellae, averaged over 30 randomized trials and lamellae removal conditions.

The online version of this article includes the following figure supplement(s) for figure 2:

**Figure supplement 1.** Distribution of parameters and conduction velocities in the single neuron model cohort.

regardless of how many segments were affected. However, when all lamellae were removed, CV slowed drastically – by 38±10% even when just 25% of the segments were demyelinated in this way, and 35±13% of APs failed. When 75% of segments lost all their lamellae, CV slowed by 72±8% and 45±13% of APs failed. Responses to demyelination sometimes varied widely across the cohort. We employed Lasso regression to identify key parameters that contributed to CV changes, since those

**Table 1.** Axon parameter ranges for Latin hypercube sampling (LHS) construction.

| Parameter | Values | |
|---|---|---|
| | **Minimum** | **Maximum** |
| Axon diameter (µm), measured at nodes | 0.5 | 1.02 |
| Node length (µm) | 0.25 | 2.02 |
| Myelinated segment length (µm) | 50 | 200 |
| Number of myelin lamellae | 5 | 20 |
| Lamella thickness (µm) | 0.013 | 0.019 |
| Scale factor for leak conductance | 0.1 | 1 |
| Scale factor for NaF maximal conductance | 0.1 | 1 |
| Scale factor for KDR maximal conductance | 0.1 | 1 |

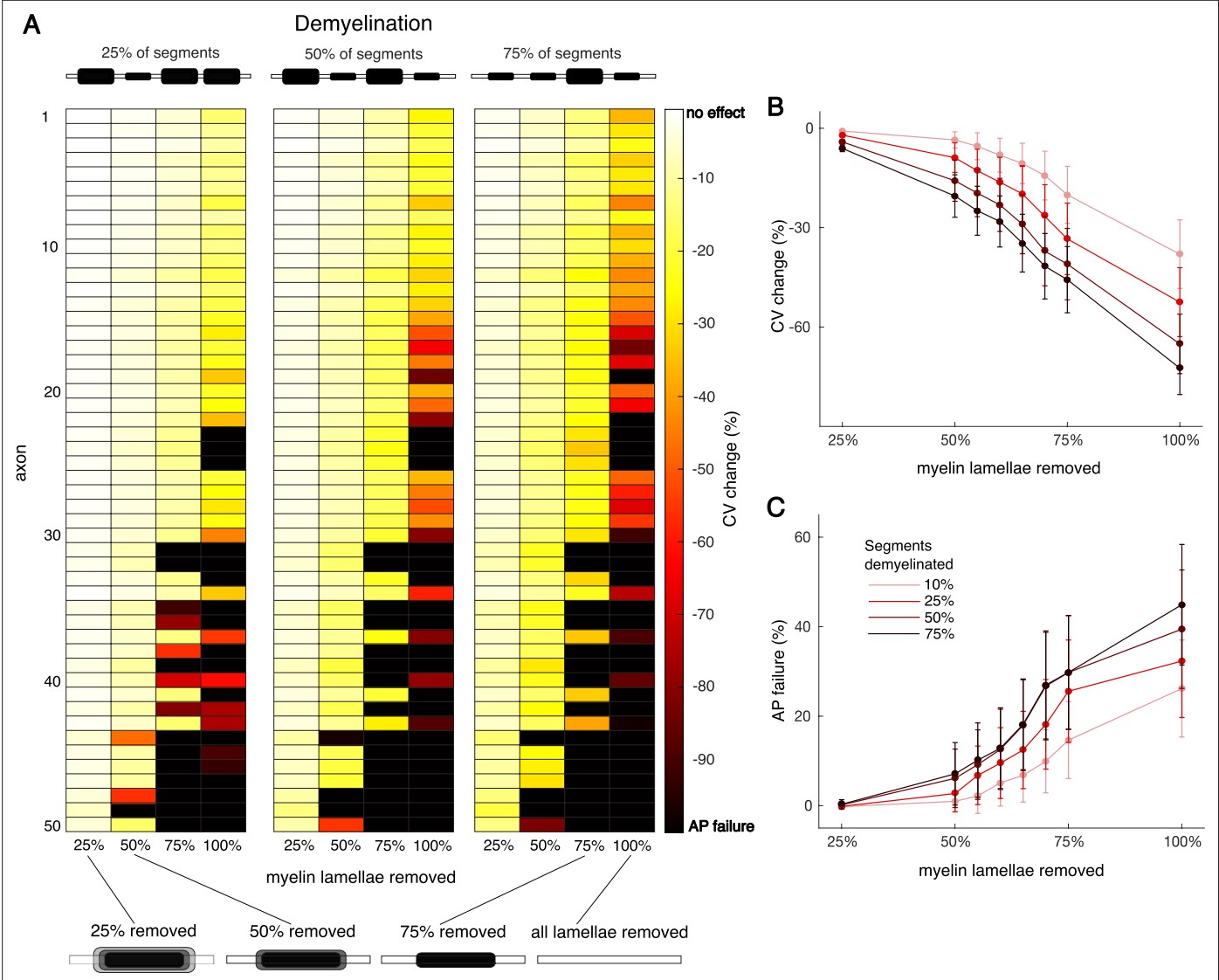

**Figure 3.** Effects of demyelination on conduction velocity (CV) and action potential (AP) failures in the single neuron model. (**A**) Heat maps showing CV change (reduction relative to the CV of the corresponding unperturbed models, measured in %) in response to select demyelination conditions across the 50 cohort axons (see Methods). Axons arranged vertically in increasing order of myelinated segment length (longest at the bottom). The three blocks from left to right show increasing numbers of demyelinated segments in each axon (25%, 50%, and 75% of segments respectively), illustrated by cartoons on top. Within each block, individual columns correspond to the percentage of myelin lamellae removed from each demyelinated segment (shown in cartoons below). Color of each box indicates the mean CV change across 30 trials of each condition, ranging from 0% (no effect) to –100% (AP failure). Overall, AP propagation was increasingly impaired with increasing levels of demyelination. Mean CV change (**B**) and percentage of AP failures (**C**) versus the percentage of lamellae removed for all demyelination conditions simulated. Colors represent the percentages of segments demyelinated, from 10% (light red) to 75% (black). Error bars represent mean ± SEM, averaged across all cohort axons (n=50) and 30 trials each.

The online version of this article includes the following figure supplement(s) for figure 3:

**Figure supplement 1.** Statistical analysis of parameters contributing to conduction velocity (CV) changes after demyelination and remyelination.

changes preceded AP failures (*Figure 3—figure supplement 1A and C*). Five of the 12 parameters analyzed contributed to CV changes during demyelination. Among them, myelinated segment length had the largest magnitude with a negative weight indicating that models with longer myelinated segments show more CV slowing in response to a given demyelination perturbation. Scale factors for leak and sodium conductance, axoplasm resistance, and tight junction resistance also controlled CV changes during demyelination.

## Remyelination leads to partial recovery from CV slowing and AP failures

We next examined the extent to which remyelination with shorter and thinner segments, occurring after demyelination, restored axonal AP propagation (*Figure 4*). We first assumed that affected segments had been previously completely demyelinated, i.e., losing all their lamellae (see Methods; *Figure 4A–C*). Most remyelinated models showed CV recovery from 0% to 100%, except for a few cases in which CV increased relative to the unperturbed models (*Figure 4—figure supplement 1A*; see Discussion). The CV recovered more as both the remyelination and the lamellae restoration percentages increased. When all demyelinated segments were subsequently remyelinated with sufficient lamellae – and none of the perturbed segments were bare – the CV recovered substantially and almost no AP failed (*Figure 4B and C*). The initial fraction of demyelination also affected CV recovery, but in a more subtle way. When all demyelinated segments were remyelinated, there was a *positive* relationship between the initial demyelination rate and the CV recovery: CV recovered more when 75% of the segments were demyelinated (*Figure 4B*, black lines) than when only 25% were affected. This finding is consistent with observations of *Scurfield and Latimer, 2018*, in which axons with more transitions between long (unperturbed) and short (remyelinated) segments had slower CV (*Figure 4—figure supplement 2*). When incomplete remyelination left some segments bare (*Figure 4B*, colored lines), there was a *negative* relationship between the initial amount of demyelination and CV recovery: axons with more bare segments had reduced electrical insulation, and therefore recovered less.

We also simulated remyelination after a milder partial demyelination, where affected segments initially lost only half their lamellae (*Figure 4D*). Overall trends were similar to, but less severe than, those for the complete demyelination case (*Figure 4E and F*; *Figure 4—figure supplement 1B*). The variability in CV recovery across different remyelination conditions and across the model cohort was similar. There were also fewer AP failures under partial demyelination conditions, relative to the corresponding complete demyelination cases (*Figure 4C* vs. *Figure 4F*).

Results for the percentage of AP failures (*Figure 4C and F*) were consistent with those for CV recovery. Remyelinating all previously demyelinated segments, even adding just 10% of lamellae, brought AP failure rates down to 14.6±5.1%. Remyelinating all affected segments with 75% of lamellae (the maximal amount of remyelination) nearly eliminated AP failures (1.8±1.1%). Incomplete remyelination, where some segments were still demyelinated, still had relatively high AP failure rates. For example, when one eighth of segments were remyelinated with the maximal amount of lamellae and one eighth were left bare, 25.7±11.5% of APs failed across the cohort (*Figure 4C*, red dashed line and arrow). AP failure rates were slightly lower when starting with partial demyelination: 10.6±7.6% of APs failed in the analogous paradigm (*Figure 4F*, red dashed line and arrow). In short, combinations of demyelinated and remyelinated segments often led to sizable CV delays and AP failures. Applying Lasso regression to CV recovery after remyelination (*Figure 3—figure supplement 1C and D*) found 10 of the 12 parameters contributed significantly; only myelin length and axoplasm resistance were omitted. Comparing these parameters with empirical data, when available, may help estimate the severity of CV delays and AP failures in the cortex of aging rhesus monkeys.

## AP failures impair performance in a neural network model of working memory

Equipped with the quantification of impaired AP transmission due to myelin alterations in the single neuron model, we next elucidated how these impairments affected neural circuit function in PFC. We focused on spatial working memory because the underlying neural network mechanisms have been studied and modeled in detail (e.g. *Compte et al., 2000*; *Hansel and Mato, 2013*; *Ibañez et al., 2019*). We built on a previous spiking neural network model that accounts for many experimental findings (*Hansel and Mato, 2013*). It consists of 16,000 excitatory and 4000 inhibitory integrate-and-fire neurons (see Methods). Neurons are coupled through excitatory synapses with AMPA and NMDA receptors, and inhibitory synapses with GABA receptors. Recurrent excitatory synapses are facilitating, as has been empirically observed in PFC (*Hempel et al., 2000*; *Wang et al., 2006*), which promotes robust and reliable persistent activity despite spatial heterogeneities in the connectivity or in the intrinsic properties of the neurons.

We first simulated the classical oculomotor delayed response task (DRT, *Figure 5A*) in a cohort of 10 young (control) networks with different random connectivities and intact AP transmission (see

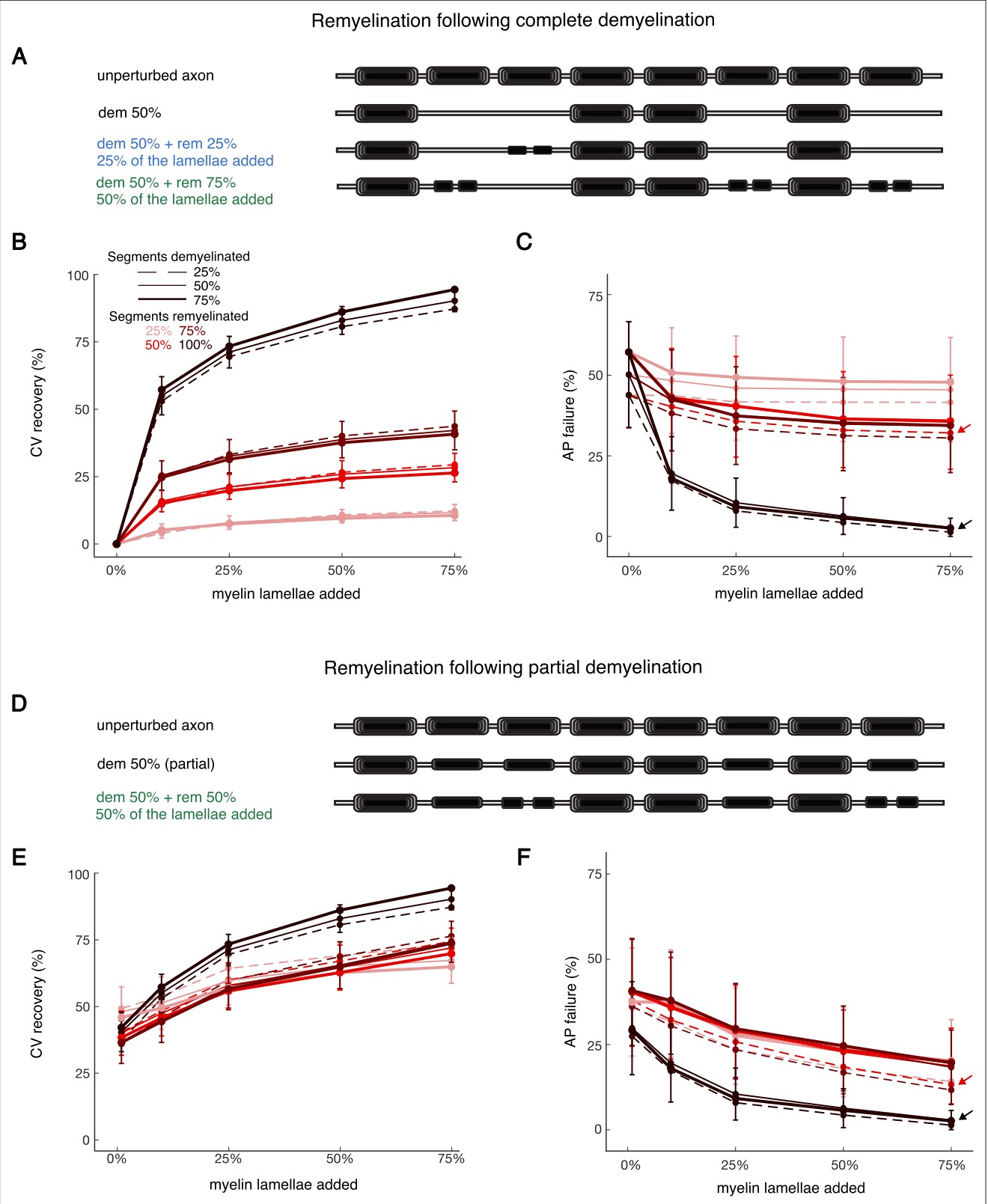

**Figure 4.** Conduction velocity (CV) recovery in response to remyelination. (**A**) Cartoons illustrating representative remyelination conditions after select segments were completely demyelinated. Top row shows an unperturbed axon with eight myelinated segments. Second row: 50% of segments are completely demyelinated. Third row: 25% of the demyelinated segments in second row (one in total) are remyelinated with two shorter segments, each with 25% of lamellae restored. Fourth row: 75% of the demyelinated segments in the second row (three in total) are remyelinated with two

*Figure 4 continued on next page*

*Figure 4 continued*

shorter segments, each with 50% of lamellae restored. Mean CV recovery (**B**) and percentage of action potential (AP) failures (**C**) versus the percentage of lamellae restored for all simulated remyelination conditions after complete demyelination. (**D**) Cartoons illustrating representative remyelination conditions after partial demyelination. Top row shows an unperturbed axon with eight myelinated segments. Second row: 50% of segments are partially demyelinated (with 50% of lamellae removed). Third row: 50% of the demyelinated segments in second row (two in total) are remyelinated with two shorter segments, each with 50% of lamellae restored. Mean CV recovery (**E**) and percentage of AP failures (**F**) versus the percentage of lamellae restored for all simulated remyelination conditions after partial demyelination. CV recovery in both cases (**B and E**) was calculated with respect to the CV change for the complete demyelination (see Methods). In panels (**B**, **C**, **E**, and **F**), the x-axis refers to the percentage of myelin lamellae restored relative to unperturbed segments, starting at 0% (no remyelination). Line styles represent the percentage of segments initially demyelinated, from 25% (dashed) to 75% (thick solid). Colors represent the extent of remyelination, from 25% (light red) to 100% (black). Shown are mean values, averaged across all cohort axons (n=50) and 30 trials each. For readability, error bars (representing ± SEM) are shown only for the condition of 50% demyelination of segments.

The online version of this article includes the following figure supplement(s) for figure 4:

**Figure supplement 1.** Conduction velocity (CV) recovery in response to remyelination across the model cohort.

**Figure supplement 2.** Transitions between myelinated segments of dissimilar lengths affect response to perturbations.

Methods). For appropriate levels of excitation and inhibition, a localized activity bump forms during the cue period, and this bump persists through the delay period. The center of the bump encodes the remembered spatial location (*Figure 5B, i*; *Compte et al., 2000*; *Hansel and Mato, 2013*). Successful trials require a sufficiently strong activity bump throughout the delay period, quantified by the memory strength (*Figure 5C*, blue line). If the memory strength decreases over time (e.g. caused by the demyelination/remyelination conditions discussed below) the memory duration – the period during which the network can retain the stimulus– becomes limited (*Figure 5C*). Moreover, due to random fluctuations, the activity bump diffuses along the network during the delay period. This leads to trial-to-trial variability in the cue position read out from the network activity, modeling the variability of recalled spatial locations observed empirically (*Figure 5B*, right panels; *Wimmer et al., 2014*). This memory diffusion increases with the delay duration, consistent with decreasing working memory precision observed experimentally (*Figure 5D*; *Funahashi et al., 1989*). The bump movement during the delay also has a directed component, i.e., a systematic bias clockwise or counterclockwise away from the cue location, caused by heterogeneities in the network connectivity (*Figure 5B*, right panels and *Figure 5E*; *Hansel and Mato, 2013*). This memory drift is a possible explanation for delay-dependent systematic biases in working memory observed experimentally (see Discussion). However, because of their established relationship with working memory performance, in the following we focus on memory duration, corresponding to complete forgetting, and memory diffusion, corresponding to working memory precision.

To investigate how myelin alterations affect working memory maintenance, we explored in the network model the same demyelination and remyelination conditions as we did in the single neuron model. Because our network model consists of point neurons (i.e. without detailed axons), we incorporated CV slowing as an effective increase in synaptic transmission delays (see Methods). To simulate AP failures, we adjusted the AP failure rate to the values given by the single neuron model, by creating a probabilistic model of spike transmission from the excitatory presynaptic neurons to both the excitatory and inhibitory postsynaptic neurons (see Methods). We found that propagation delays, even larger than those quantified with the single neuron model, had no effect on the network performance. Only unrealistically long delays led to a slight decrease in performance (*Figure 5—figure supplement 2*). This was as expected because, by design, the network operates in an asynchronous state with irregular neural activity in which the timing of individual spikes does not affect network function. AP failures, on the other hand, did have a large impact. For AP failure probabilities matched to the distribution of AP failure probabilities across the cohort of 50 single neurons above, we observed a decay of the activity bump (i.e. reduced memory strength over time) in the network model. This can ultimately lead to extinction of the activity bump during the delay period, representing complete forgetting of the remembered stimulus (reduced memory duration; *Figure 5B, ii and iii*, *Figure 5C*, and *Figure 6—figure supplement 1A*). For a mild reduction of the bump amplitude, memory duration was not affected (*Figure 5C*, purple line) but memory diffusion increased compared to the control network (*Figure 5B, iii* and *Figure 5D*). This increase in memory diffusion is consistent with mathematical

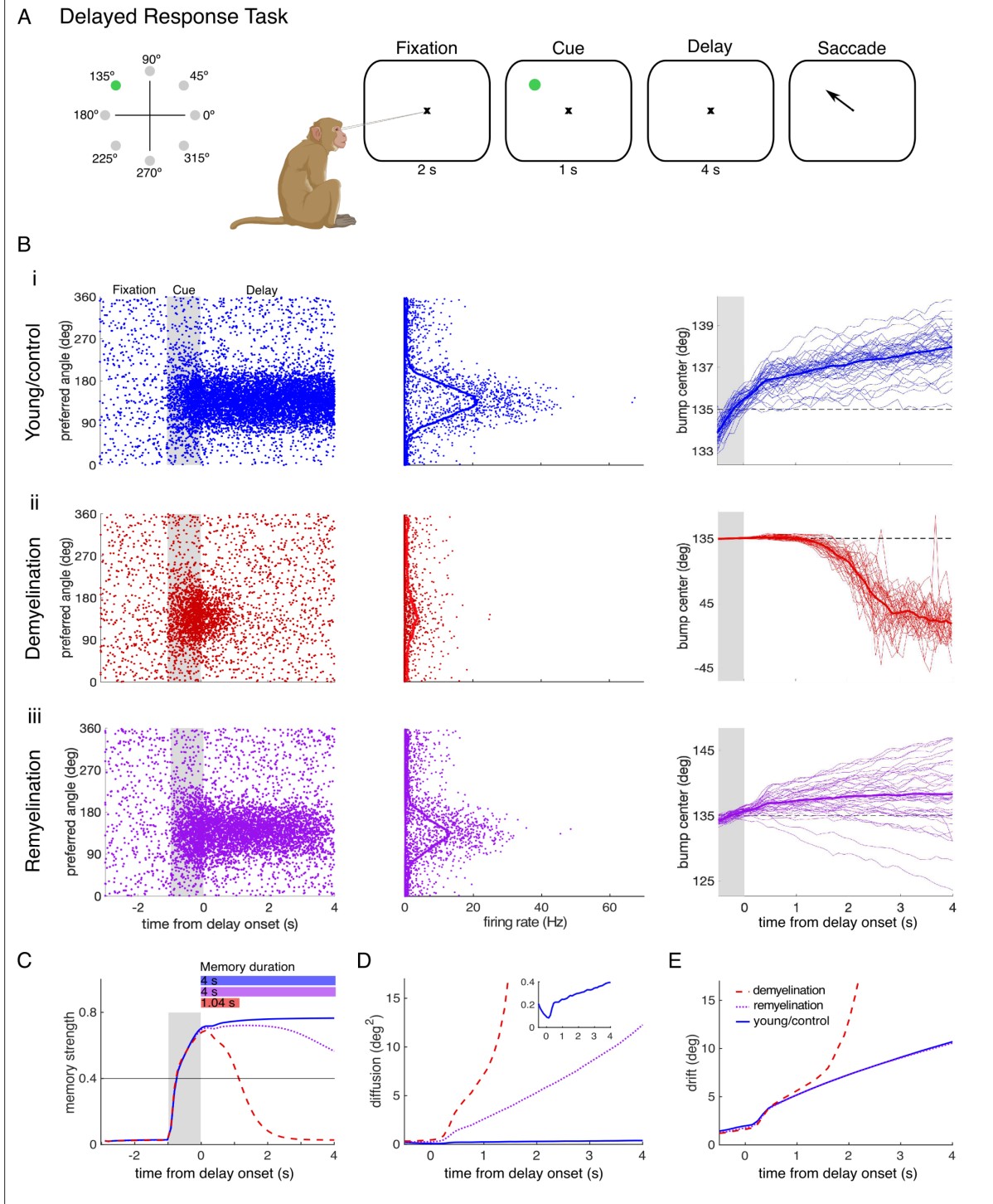

**Figure 5.** Action potential (AP) failures impair working memory performance in a spiking neural network model. (**A**) Schematic of the delayed response task. Subjects fixate at the center of a computer screen and need to remember a cue stimulus, presented at one out of eight locations throughout the delay period, before indicating the remembered location with an eye movement. (**B**) Excitatory neuron activity for a cue stimulus presented at 135° of an (**i**) unperturbed control network, (ii) a network with demyelination, and (iii) a network with remyelination. Left: Single-trial raster plot showing the activity for each neuron (labeled by its preferred direction) during the precue (fixation), cue and delay periods of the task. The cue period is indicated by the gray shading. Middle: Average spike counts of the excitatory neurons during the delay period. The points show average spike rates of individual neurons and the solid line the average over 500 nearby neurons. Right: Trajectory of the bump center (i.e. the remembered cue location) read out from the neural activity across the cue and delay periods using a population vector (see Methods). Thin lines correspond to individual trials and the solid line to the trial average. (ii) Shows the effect of AP failure probabilities corresponding to demyelination of 25% of the myelinated segments by removing 75%

*Figure 5 continued on next page*

*Figure 5 continued*

of the myelin lamellae. (iii) Corresponds to AP failure probabilities for remyelination of 50% of the demyelinated segments by adding 75% of the myelin lamellae back, after previous partial demyelination of 25% of the segments. (**C**) Memory strength as a function of time and corresponding memory duration (horizontal bars; memory strength ≥0.4; see Methods). (**D**) Working memory diffusion (trial-to-trial variability of bump center) during the cue and delay periods. The inset shows a close-up of the diffusion for control networks. A similar increase of working memory diffusion with demyelination is also observed in networks with overall higher diffusion (*Figure 5—figure supplement 1*). When demyelination is restricted to a part of the network, diffusion only increases in the perturbed zone (*Figure 5—figure supplement 3*). (**E**) Working memory drift (systematic memory errors). Note that the remyelination curve (purple dotted line) in (**E**) superimposes the young curve (blue solid line). The red dashed line represents the demyelination case. The performance measures in (**C–E**) were obtained by averaging across 280 trials and 10 networks, either control (**B, i**) or perturbed (**B**, ii–iii).

The online version of this article includes the following figure supplement(s) for figure 5:

**Figure supplement 1.** Increased working memory diffusion in spiking networks with spatially correlated background inputs.

**Figure supplement 2.** Effect of propagation delays on control and perturbed networks.

**Figure supplement 3.** Effect of spatially heterogeneous demyelination of the model neurons according to their preferred angle.

**Figure supplement 4.** Action potential (AP) failures impair working memory performance in a network model with activity-silent memory traces.

**Figure supplement 5.** Effect of propagation delays on control and perturbed activity-silent network models.

---

analysis of firing rate models showing that the bump diffusion depends inversely on the squared bump amplitude (*Kilpatrick and Ermentrout, 2013*; *Esnaola-Acebes et al., 2022*).

## Impact of demyelination and remyelination on working memory

We then systematically characterized changes in memory duration and memory diffusion by comparing working memory performance in the cohort of 10 control networks with the performance of those networks perturbed corresponding to varying degrees of demyelination and remyelination. In each of the 10 networks, we set the AP failure rate of the excitatory neurons according to the distribution of failure probabilities of the neurons in the single neuron cohort for the given demyelination or remyelination condition. Thus, we took into account the heterogeneity of demyelination and remyelination effects from our single neuron cohort (*Figure 3A*; *Figure 4—figure supplement 1*). Note that this heterogeneity originates from differences in axon properties, but probabilities of failure for all neurons in the network correspond to the same degree of demyelination (*Figure 6*). We will also consider networks that contain different combinations of axons with either intact or perturbed myelin (*Figure 7* and *Figure 8*).

### Demyelination impairs working memory performance compared to control networks

We found that the memory duration was not affected when removing up to 55% of myelin lamellae per myelinated segment, regardless of the percentage of axonal segments that were altered along an axon (*Figure 6A*, left panel). However, when between 55% and 75% of the myelin lamellae were removed, the memory duration began to decrease, depending on the percentage of axonal segments that were demyelinated. In this case, increasing the percentage of demyelinated segments from 10% to 50% led to a progressive impairment, whereas an increase to 75% of the segments did not impair memory duration further. Finally, in cases where 100% of the myelin lamellae were removed, the memory duration dropped to ≤1 s, regardless of the percentage of segments that were demyelinated. In a similar trend, memory diffusion increased, i.e., working memory became less precise starting when removing between 25% and 50% of the myelin lamellae (*Figure 6B*, left panel). Again, we found a progressive impairment, depending on the percentage of myelin lamellae removed and the percentage of myelinated segments affected, with a ceiling effect when more than 50% of segments were demyelinated.

### Complete remyelination recovers network function

We observed that remyelination of all previously demyelinated segments (100%), independently of the degree of demyelination, recovered the memory duration to the control networks-like performance (*Figure 6A*, middle and right panels; black lines). However, working memory precision is not fully recovered in all these cases, indicated by an increase in the diffusion constant (*Figure 6B*). The performance of control networks was completely recovered only when 75% of the myelin lamellae were added back to

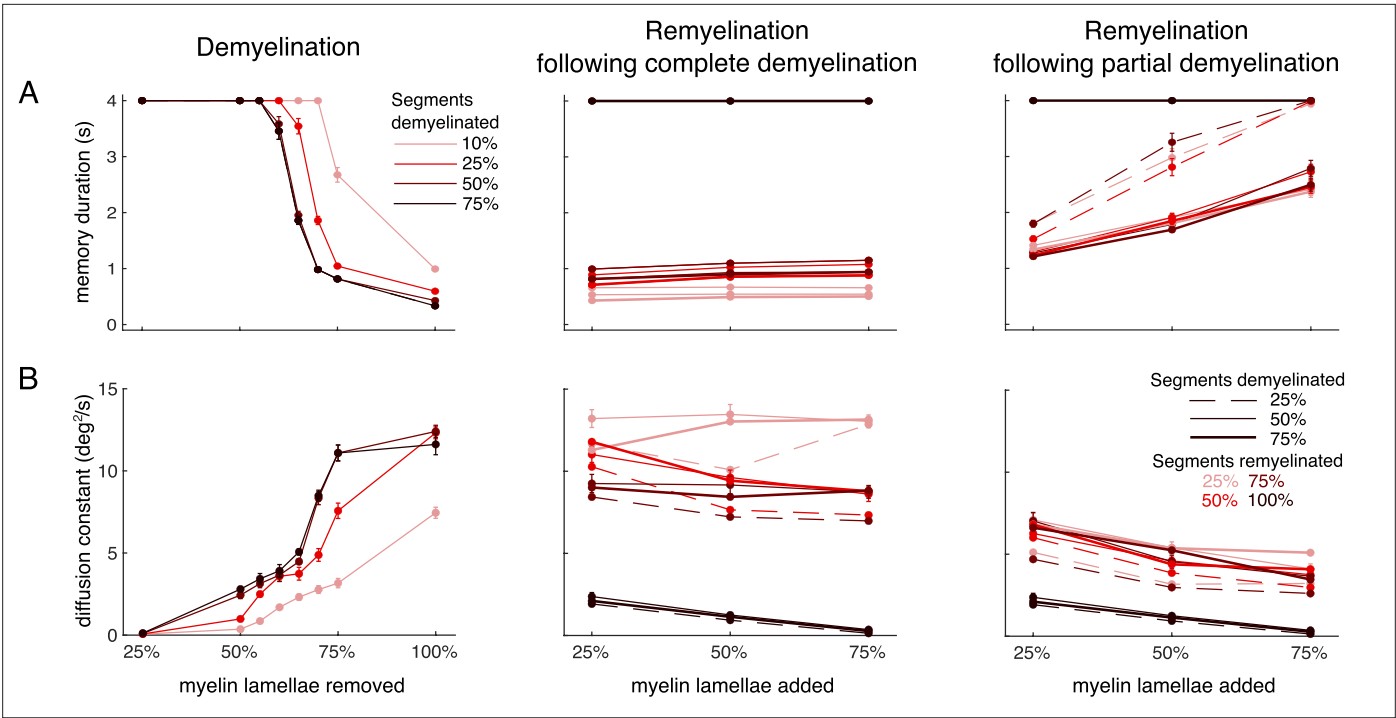

**Figure 6.** Working memory function in the network model is impaired by demyelination and recovered by sufficient remyelination. (**A**) Memory duration and (**B**) diffusion constant for simulations of the delayed response task, as in **Figure 5**, for a systematic exploration of the effect of action potential (AP) failure probabilities corresponding to the different demyelination and remyelination conditions explored with the single neuron model. Left panel: Demyelination, realized by removing a fraction of myelin lamellae from a fraction of myelinated segments. Middle panel: Remyelination with two shorter and thinner myelin sheaths, with a node in between, of the previously completely demyelinated segments. Right panel: Same as the middle panel but for partial demyelination (removal of 50% of the myelin lamellae) rather than complete demyelination. In all cases, the performance measures were obtained by averaging across the 10 perturbed cohort networks and the 280 trials simulated for each network. The average memory duration for the 10 unperturbed, control networks in the cohort (averaged across 280 trials) was 4 s, and the average diffusion constant was 0.064 (both values corresponding to the case of 0% of myelin lamellae removed in the left panels of (**A**) and (**B**), respectively; not shown). Error bars represent mean ± SEM, averaged across all networks and trials.

The online version of this article includes the following figure supplement(s) for figure 6:

**Figure supplement 1.** Memory strength decreases for different degrees of demyelination and remyelination.

**Figure supplement 2.** Increase of memory drift for different degrees of demyelination and remyelination.

the remyelinated segments. Thus, despite the new shorter and thinner myelin sheaths compared to the original intact ones, complete remyelination is able to recover control, unperturbed network function.

## Incomplete remyelination leads to partial recovery

We studied the effect of incomplete remyelination (remyelination of 25–75% of previously demyelinated segments) after both complete and partial demyelination (**Figure 6A and B**, middle and right panels, respectively). When we remyelinated between 25% and 75% of the previously completely demyelinated segments, we did not observe a significant recovery of the memory duration (**Figure 6A**, middle panel; memory duration ≲ 1 s in all cases). Memory diffusion was partly recovered compared to the complete demyelination case (compare left and middle panels in **Figure 6B**) when 50–75% of the demyelinated segments were remyelinated, but it remained far from the performance of the control network. In sum, incomplete remyelination was unable to restore network function when bare axon (completely demyelinated) segments were present. However, when we remyelinated between 25% and 75% of the previously only partially demyelinated segments, both memory duration and memory diffusion were restored closer to the values of the control networks (**Figure 6A and B**, right panel).

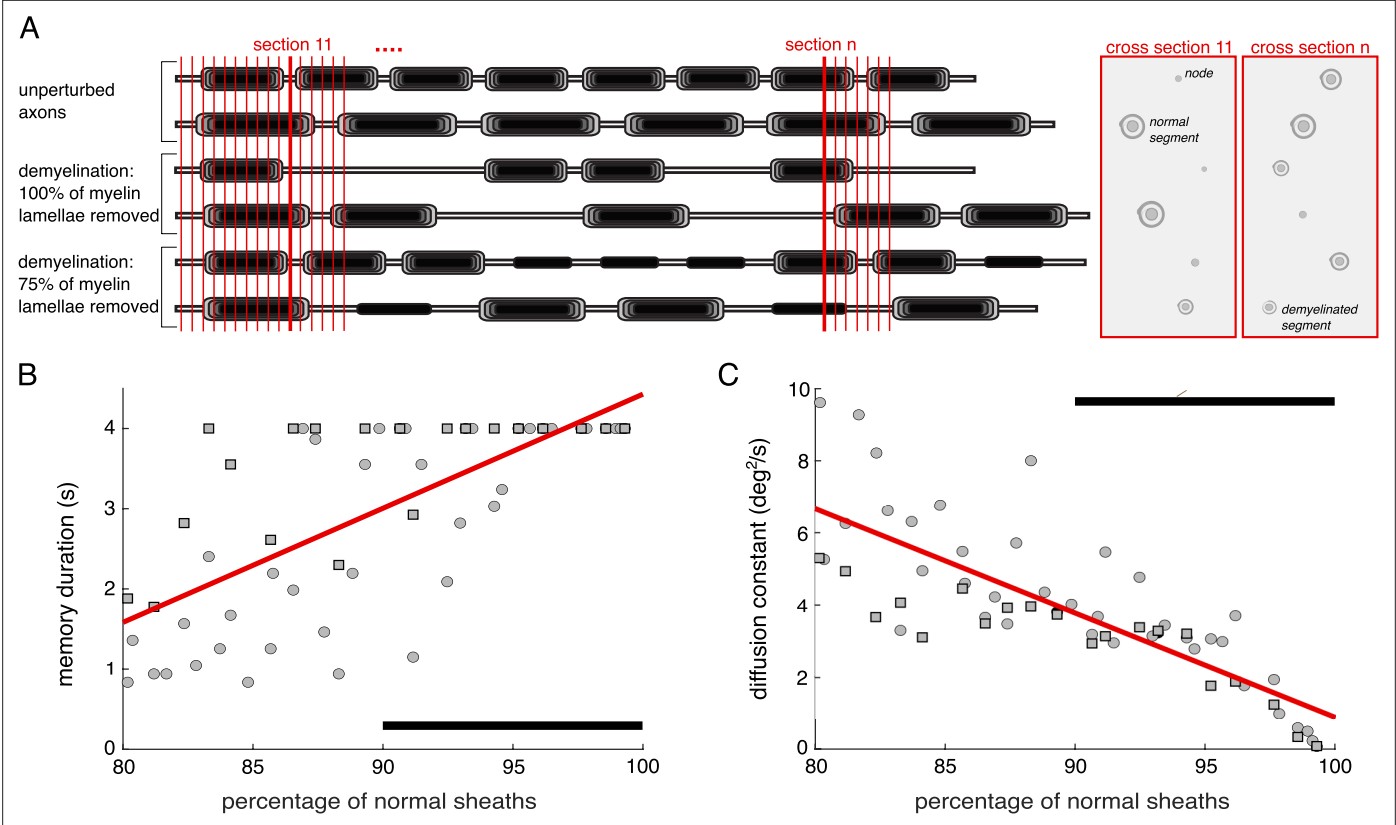

**Figure 7.** Reduced normal myelin is associated with decreased working memory performance in the network model. (**A**) Schematic of the quantification of unperturbed, normal myelin sheaths in groups of neurons containing intact and demyelinated axons with different proportions of demyelinated segments (see Methods). Vertical red lines indicate cross-sectional planes that mimic electron microscopy images capturing cross sections of different axonal parts. (**B**) Memory duration and (**C**) diffusion constant vs. the percentage of normal myelin sheaths. Linear regressions show significant positive correlations in both cases (memory duration: $r=0.703$, $p=3.86 \times 10^{-10}$; diffusion constant: $r=-0.802$, $p=1.26 \times 10^{-14}$). Circles: All the demyelinated segments in the perturbed axons in the groups were bare segments (all myelin lamellae removed). Squares: All the demyelinated segments in the perturbed axons had 75% of the myelin lamellae removed. Black horizontal bars indicate the percentage of normal sheaths observed in electron microscopy images from young and aged rhesus monkeys dorsolateral prefrontal cortex (dlPFC) (*Peters and Sethares, 2002*).

## Alternative working memory mechanisms

Working memory in our neural network is maintained in an attractor state with persistent neural activity (*Compte et al., 2000*; *Hansel and Mato, 2013*). Other mechanisms have been proposed, including that working memory maintenance may rely on activity-silent memory traces (*Mongillo et al., 2008*; *Stokes, 2015*; *Barbosa et al., 2020*). In activity-silent models, a slowly decaying transient of synaptic efficacy preserves information without the need for persistent ongoing activity. We implemented an activity-silent model, to our knowledge the first one for continuous spatial locations, and tested how working memory performance is affected by AP failures and propagation delays. We found that AP failures corresponding to demyelination caused working memory errors qualitatively similar to the delay-active network (*Figure 5—figure supplement 4*). On the other hand, increasing propagation delays did not lead to additional working memory errors, unless we include unrealistically high values (uniform distribution in the range of 0–100 ms; *Figure 5—figure supplement 5*). These results are qualitatively similar to the delay-active network model. Thus, our main findings do not critically depend on the exact working memory mechanism (active vs. activity-silent).

## Simulated heterogenous myelin alterations match empirical data

Up to this point we have studied network models with AP failure probabilities corresponding to a single degree of myelin alterations (i.e. with all excitatory neurons in the network having AP failure rates matched to those of the single neuron cohort for one particular demyelination or remyelination condition). Next, we sought to reveal the effect on working memory performance of more biologically

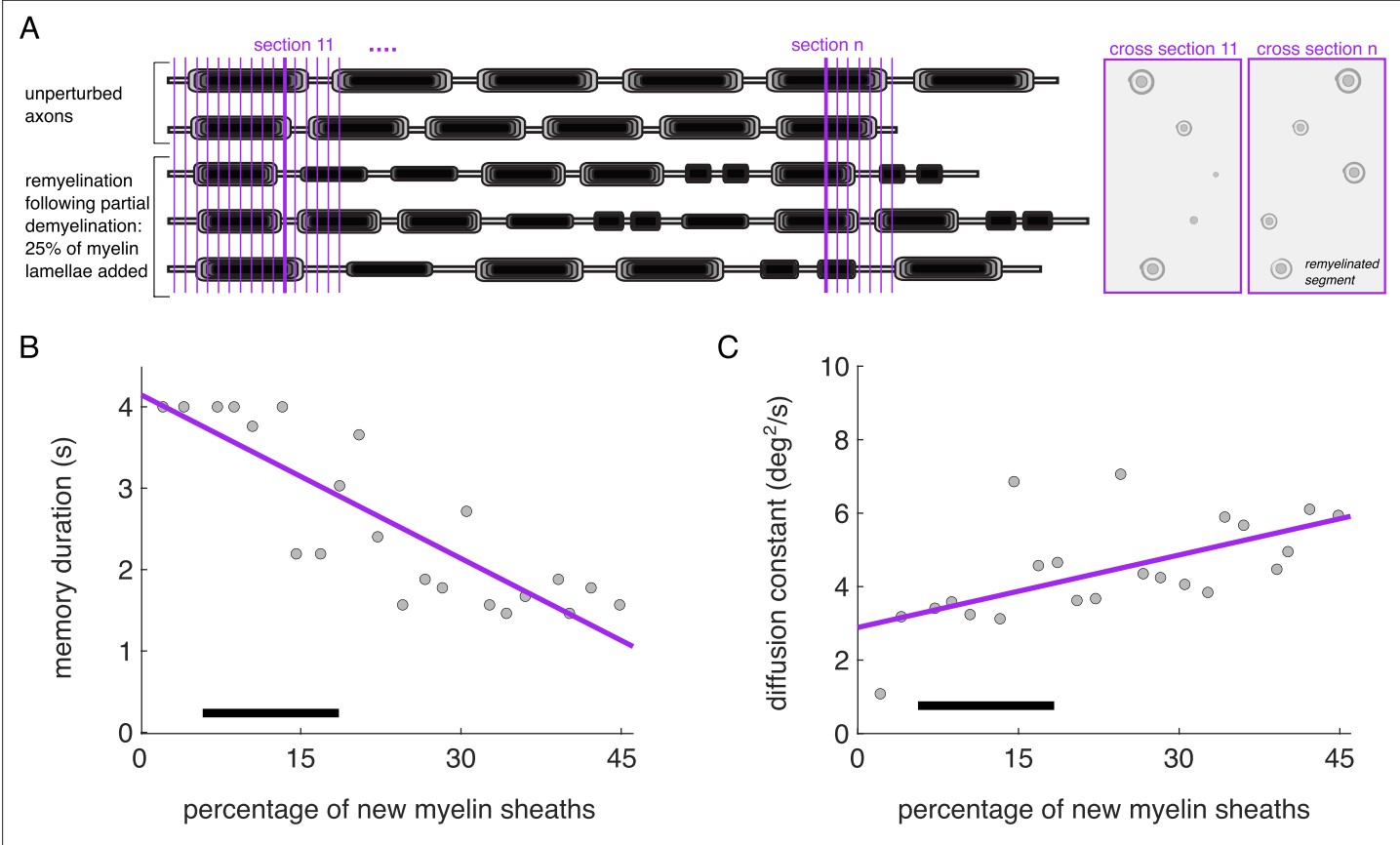

**Figure 8.** A higher proportion of new myelin sheaths impairs working memory in the network model. (**A**) Schematic of the quantification of new myelin sheaths in groups of neurons containing intact and partly remyelinated axons. Vertical purple lines indicate cross-sectional planes that model electron microscopy images capturing cross sections of different axonal parts. (**B**) Memory duration and (**C**) diffusion constant vs the percentage of new myelin sheaths. Linear regressions show significant negative correlations in both cases (memory duration: $r=-0.852$, $p=4.92 \times 10^{-7}$; diffusion constant: $r=0.607$, $p=0.003$). The remyelinated axons in the groups have different proportions of segments remyelinated after partial demyelination, by adding 25% of the myelin lamellae back. Black horizontal bars indicate the percentage of paranodal profiles observed in electron microscopy images from young and aged rhesus monkeys dorsolateral prefrontal cortex (dlPFC) (*Peters and Sethares, 2003*).

realistic network models, where excitatory neurons in the networks were perturbed according to a combination of different demyelination or remyelination conditions. That is, we simulated networks with excitatory neurons having AP failure probabilities matched to both neuronal axons with intact and with altered myelin sheaths in different degrees, as likely occurs in the aging brain (*Figure 1*).

### Fewer normal myelin sheaths lead to decreased network performance

We ran network model simulations combining AP failure probabilities corresponding to groups of neurons containing either intact axons or axons presenting different degrees of demyelination (*Figure 7A*; Methods). Quantifying the average degree of demyelination in each simulated network allowed us to predict working memory deficits for a degree of demyelination that is within the empirically observed range of 90–100% normal myelin (*Peters and Sethares, 2002*). We observed that the performance was impaired – memory duration significantly decreased (*Figure 7B*), and memory diffusion significantly increased (*Figure 7C*) – when the percentage of normal sheaths decreased. These results are consistent with an experimentally observed increased cognitive impairment in various learning and working memory tasks (including the delayed recognition span task, a spatial working memory task) with an age-related decrease of the percentage of normal sheaths in dlPFC of rhesus monkeys (*Peters and Sethares, 2002*). Importantly, our results indicate that myelin alterations alone can account for significant working memory impairment, pointing to demyelination as a key factor in age-related working memory decline.

## Shorter and thinner myelinated segments impair working memory

To predict the effects of remyelination on working memory for the empirically observed range of axon remyelination, we simulated network models that contained a combination of model neurons with intact axons and with axons containing different proportions of remyelination (*Figure 8A*; Methods). Empirical studies have found an age-dependent increase in the percentage of paranodal profiles, indicative of more and shorter myelin sheaths, from 5% in young monkeys to 17% in aged monkeys (*Peters and Sethares, 2003*). Here, we used different proportions of incomplete remyelination as studied in *Figure 6* (axons with either 25%, 50%, or 75% of their segments remyelinated), and we quantified the overall percentage of new myelin sheaths in the networks. We simulated networks with up to a 45% of new myelin sheaths and we observed that performance was significantly impaired compared to the young, control networks with intact myelin: memory duration decreased to almost 1 s (*Figure 8B*) and diffusion constant increased (*Figure 8C*). This is consistent with empirical findings showing that cognitive impairment for a cohort of 18 young and aged rhesus monkeys increased with an increase of the percentage of paranodal profiles in the dlPFC (*Peters and Sethares, 2003*). Therefore, both demyelination and incomplete remyelination lead to impaired performance in our networks, compared to networks with intact myelin sheaths.

## Discussion

This multi-level computational study explored how age-related myelin degradation (demyelination) and remyelination affect AP propagation in individual axons and working memory precision in spiking neural networks. We found that these myelin changes lead to AP failures which, if uncompensated by other factors, predict working memory decline with aging.

### Myelin changes affect AP propagation in a cohort of model neurons

The novelty of our neuron model lies in its systematic exploration of a combination of different myelin perturbation types known to occur in myelin dystrophies, across a wide range of biologically feasible models. Our single neuron model assumed that age-related myelin dystrophies (e.g. *Figure 1*) alter the insulative properties of lamellae analogously to demyelination, and examined interactions between demyelination and remyelination. Past studies of myelin dystrophy examined how either demyelination or remyelination of all segments affected AP propagation for a few representative axon morphologies. For example, *Scurfield and Latimer, 2018*, explored how remyelination affected CV delays, finding that axons with more transitions between long and short myelinated segments had slower CV (*Figure 4—figure supplement 2*), and was first to explore how remyelination interacts with tight junctions. However, their study did not couple remyelination and demyelination together or examine AP failures. Other basic findings from our single neuron cohort are consistent with past modeling studies, including that demyelination caused CV slowing and eventual AP failures (*Stephanova et al., 2005*; *Stephanova and Daskalova, 2008*; *Naud and Longtin, 2019*), and, separately, that remyelination with shorter and thinner myelinated segments led to CV slowing (*Lasiene et al., 2008*; *Powers et al., 2012*; *Scurfield and Latimer, 2018*). However, by assuming that some previously demyelinated segments were remyelinated while others were not, we found that models could have much higher AP failure rates than previously reported. Such a scenario, in which individual axons have some segments that are normal, some demyelinated, and some remyelinated, is likely to occur. We also found a few neurons in our cohort showing a CV *increase* after remyelination, which has not generally been reported before and is likely due to an interplay between ion channels in the new nodes and altered electrotonic lengths in the perturbed myelinated segments (e.g. *Moore et al., 1978*; *Naud and Longtin, 2019*).

Since our single neuron cohort sampled a wide range of parameter space, we used Lasso regression to identify which of the complex, interacting parameters contributed most to CV delays (which preceded AP failures). Parameters including axon diameter, node length, length of myelinated segments, and nodal ion channel densities predicted how our models responded to demyelination and remyelination; these findings are consistent with past modeling studies over more limited parameter ranges (e.g. *Goldman and Albus, 1968*; *Moore et al., 1978*; *Babbs and Shi, 2013*; *Young et al., 2013*; *Schmidt and Knösche, 2019*). Better empirical measurements of these parameters in monkey dlPFC, e.g., from three-dimensional electron microscopy studies or single neuron axon

studies combined with markers for myelin, would help predict the extent to which myelin dystrophy and remyelination along individual axons with aging affect AP propagation.

Another important feature of our multicompartment model is that it was constrained by morphological and physiological data in rhesus monkey dlPFC –an extremely valuable dataset from an animal model with many similarities to humans (*Upright and Baxter, 2021*; *Tarantal et al., 2022*). While beyond the scope of the current study, this computational infrastructure – with a detailed axon, initial segment, soma, and apical and basal dendrites – enables simultaneous investigations of signal propagation through the dendritic arbor and axon. Our model can also be extended to explore interactions between spatially localized myelin perturbations (such as those seen in multiple sclerosis) and axon collateralization (*Sengupta et al., 2023*), which would affect the distance dependence of AP failures. Integrating such results from single neuron models into network models of working memory, as we have done here, is a powerful way to connect empirical data across multiple scales.

## Myelin changes impair working memory function

With the spiking neural network model, we found that increasing probabilities of AP failure, corresponding to higher degrees of demyelination, gradually impaired working memory precision and the time during which stimulus information can be held in memory. Complete remyelination of all previously demyelinated segments restored performance to the control network level. In contrast to the strong impact of AP failures on network function, introducing propagation delays to mimic AP slowing, in the time range of the delays quantified with the single neuron model, did not have an effect (*Figure 5—figure supplement 2* and *Figure 5—figure supplement 5*). This is because the network operates in an asynchronous state in which the dynamics are primarily governed by the statistics of neuronal activity (e.g. firing rates), rather than precise spike timings (*van Vreeswijk and Sompolinsky, 1996*; *Hansel and Mato, 2013*). While highly irregular persistent activity is indeed observed in PFC during working memory tasks (*Compte et al., 2003*), at the mesoscopic level oscillations of the local field potentials and synchronization also play an important role (*Gregoriou et al., 2009*; *Liebe et al., 2012*; *Buschman et al., 2012*; *Salazar et al., 2012*). AP slowing may alter these neural oscillations and synchrony which could lead to further working memory impairment not captured by our network model. In addition to age-related changes in memory duration and precision, our network model predicts an age-related increase in systematic errors (bias) due to an increased drift of the activity bump (*Figure 6—figure supplement 2*). Moreover, if demyelination is spatially localized in a part of the network, the model predicts a repulsive bias away from the memories encoded in the affected zone (*Figure 5—figure supplement 3*). Delay-dependent systematic working memory errors have been observed in behavioral experiments (*Panichello et al., 2019*; *Bae, 2021*; *Stein et al., 2021*) and it would be interesting to test whether those biases also change with aging. In addition to the prefrontal cortex, working memory is likely sustained by interactions between several fronto-parietal brain areas (*Leavitt et al., 2017*; *Christophel et al., 2017*). Our future work will include development of large-scale models of working memory (*Mejías and Wang, 2022*), incorporating myelin alterations in local circuits and in inter-area connections.

For biologically realistic combinations of neurons with intact and demyelinated axons, our network model predicts that myelin dystrophies alone would lead to spatial working memory impairment with aging (*Figure 7*). This result supports the observation that cognitive impairment increases as the percentage of normal myelin sheaths decreases (*Peters and Sethares, 2002*). In addition, combining neurons containing either intact or incompletely remyelinated axons, we found that spatial working memory is still impaired in the context of incomplete remyelination (*Figure 8*). Our result explains two separate empirical findings. One study showed a positive correlation between cognitive impairment and a higher percentage of new, shorter and thinner, myelin sheaths (*Peters and Sethares, 2003*). In our single neuron model, insufficiently remyelinated axons with an array of long unperturbed, long thin, and new shorter and thinner segments had slower CV and more AP failures compared to unperturbed axons. This caused WM errors in the network model. A second study found that, with aging, oligodendroglia has a decreased capacity for effective maturation, remaining in a progenitor state without being able to produce new myelin. This low capacity for remyelination was correlated with spatial working memory impairment in the monkeys (*Dimovasili et al., 2023*). Our models showed that axons with demyelinated segments, or otherwise poorly insulated myelin sheaths, experience delayed and even failed AP propagation, which in turn led to working memory impairment. Therefore,

it is reasonable to assume that ineffective remyelination may lead to working memory impairment. In fact, complete remyelination of all previously demyelinated segments with sufficient myelin, with fewer transitions between long and short segments, led to full recovery of working memory function. Our findings also suggest that differences in the degree of demyelination or remyelination (*Figures 7 and 8*) may account for the cognitive variability observed across individuals with aging, which encompass individuals with both good (*successful agers*) and impaired (*unsuccessful* agers) cognitive function (*Lacreuse et al., 2005*; *Moore et al., 2006*; *Moss et al., 2007*; *Moore et al., 2017*).

## Conclusions

The multiscale modeling approach we employed here extends our prior framework for studying how changes in the aging monkey dlPFC might affect working memory (review: *Luebke et al., 2010*). Our previous work (*Ibañez et al., 2019*) modeled increased AP firing rates observed in vitro together with the loss of both excitatory and inhibitory synapses, and quantified how these alterations affected working memory performance. Aged networks which compensated the loss of excitatory and inhibitory synapses with higher firing rates in individual pyramidal neurons successfully retained memory of the DRT stimulus. In addition, networks in which we also decreased the overall excitatory drive to pyramidal neurons reproduced the lower firing rates during performance of DRT in aged monkeys, reported in *Wang et al., 2011*. Such decreased excitation could arise from AP failures induced by myelin dystrophy, as shown here. It has also been hypothesized that the hyperexcitability of dlPFC pyramidal neurons observed in vitro with aging could be a homeostatic mechanism to compensate for AP disruptions due to dystrophic myelin (*Luebke et al., 2010*). The multicompartment model used here is particularly well suited to simultaneous modeling of alterations in the soma, dendrites, synapses, and axon of individual pyramidal neurons. This can include pathological changes to the nodes of Ranvier (reviewed in *Arancibia-Carcamo and Attwell, 2014Arancibia-Carcamo and Attwell, 2014*) or to the metabolism of axons after demyelination (*Gerevich et al., 2023*), which were not modeled here. The effects of such changes can then be incorporated into our DRT model, or models of other working memory tasks.

Myelin dystrophy occurs in many neurological conditions, including multiple sclerosis, schizophrenia, bipolar disorder, autism spectrum disorders, and after traumatic brain injury (*Franklin and Ffrench-Constant, 2008*; *Takahashi et al., 2011*; *Armstrong et al., 2016*; *Gouvêa-Junqueira et al., 2020*; *Galvez-Contreras et al., 2020*; *Simkins et al., 2021*; *Valdés-Tovar et al., 2022*). As in normal aging working memory is often one of the most vulnerable cognitive functions in these conditions, especially schizophrenia or autism spectrum disorders (*Wang et al., 2017*; *Hahn et al., 2018*; *Gold and Luck, 2023*). Since we found that myelin changes alone can account for working memory impairment, our study points to myelin degradation as a key factor in working memory decline with normal aging and, perhaps, in neuropathological conditions.

## Methods

The single neuron and network models for this study are available on ModelDB (https://modeldb.science), accession number 2014821.

### Single neuron model

To simulate age-related myelin dystrophies in individual neurons, we used a biophysically detailed multicompartment model of a rhesus monkey dlPFC Layer 3 pyramidal neuron (*Rumbell et al., 2016*), executed in the NEURON simulation environment (*Carnevale and Hines, 2006*). The soma, apical, and basal dendritic arbors were constructed schematically, scaled to the overall surface area of a three-dimensional morphological reconstruction from empirical data (68 compartments total; details in *Rumbell et al., 2016*). Ion channel conductances and kinetics were fit to electrophysiological data obtained in vitro from the same neuron. In addition to passive membrane dynamics, the model included two sodium channels (fast inactivating, NaF; and non-inactivating persistent, NaP), three potassium channels (delayed rectifier, KDR; muscarinic receptor-suppressed, KM; transient inactivating A-type, KA), a high-threshold non-inactivating calcium channel (CaL), a calcium-dependent slow potassium channel (KAHP), and the hyperpolarization-activated anomalous rectifier channel (AR). Because our prime focus was quantifying alterations in AP propagation along the axon under dystrophic myelin

conditions, we augmented the axon hillock and initial segment of the *Rumbell et al., 2016*, model (5 compartments each) by attaching nodes and myelinated segments from a detailed axon model (*Gow and Devaux, 2008*; *Scurfield and Latimer, 2018*). To the initial segment we attached 101 nodes (13 compartments each) alternating with 100 myelinated segments. Each myelinated segment was bound by a group of four paranodes (5 compartments each) on either side, with an internode flanked by two juxtaparanodes (9 and 5 compartments each, respectively) sandwiched between the paranode groups (*Figure 2A*). On both ends of a myelinated segment the extreme outward paranode interfaced with the adjoining node. Segments between successive nodes were endowed with myelin lamellae (wraps), and with tight junctions between the innermost lamella and the axolemma necessary for improving insulation and accurate modeling of AP propagation in nerve fibers with diameters less than 0.9 µm as often observed in PFC. Axon compartments included passive membrane dynamics as well as the NaF and KDR channels in the nodes. Simulations used a fixed 0.025 ms time step.

As done during in vitro electrophysiological experiments (*Chang et al., 2005*; *Ibañez et al., 2019*) and past modeling studies (*Coskren et al., 2015*; *Rumbell et al., 2016*), we first applied a holding current to stabilize the somatic membrane potential at –70 mV, then injected a current step into the somatic compartment for 2 s. We recorded the somatic firing rate, as well as the CV of APs propagating from the first node after the initial segment (proximal to the soma) and the penultimate node (at the distal end), and the percentage of APs that failed. The CV changes in response to myelin alterations were relatively insensitive to variations in the magnitude of suprathreshold somatic current steps (*Figure 2—figure supplement 1C*), and whether the current was constant or included Gaussian noise. Therefore, here we quantified CV changes and AP failures from responses to constant +380 pA current steps only.

## Building a cohort of control model neurons

Recognizing that the effects of myelin dystrophy may depend on the physical dimensions of an axon, we constructed a cohort of neuron models that were consistent with empirical measurements of axonal morphology in young adult rhesus monkeys. To form this 'control' model cohort, we identified five parameters of axonal morphology which have been measured empirically in rhesus monkey cortex: axon diameter (*Peters et al., 2001*; *Bowley et al., 2010*; DL Rosene et al., unpublished observations), node length (*Peters and Sethares, 2003*), length of myelinated segments (*Waxman, 1980*), number of myelin lamellae (*Peters and Sethares, 2003*), and thickness of each lamella (*Moore et al., 1978*; *Peters and Sethares, 2003*). We also defined three scale factors defining the ratio of leak, NaF, and KDR conductances in our nodes relative to those of the *Scurfield and Latimer, 2018*, model, assuming an upper bound of 1 for each. This gave a total of eight parameters with associated feasible parameter ranges that we varied (*Table 1*); other parameters including tight junction resistivity, myelin resistivity, and myelin capacitance were held fixed at values from *Scurfield and Latimer, 2018*. We then used the space-filling Latin hypercube sampling design (*Morris and Mitchell, 1995*; *Rumbell et al., 2016*; *Ibañez et al., 2019*) to identify a set of 1600 points in parameter space that maximized the minimum distance between all pairs of points. We simulated each model specified by these points, and had several selection criteria for identifying biologically plausible models. First, we constrained somatic firing rates within ranges observed empirically in Layer 3 pyramidal neurons of rhesus monkey dlPFC: firing 13–16 Hz in response to the +380 pA current step and silent when no current was injected (*Chang et al., 2005*; *Ibañez et al., 2019*). We also required CVs of 0.3–0.8 m/s (rhesus monkey corpus callosum, DL Rosene et al., unpublished observations), and ensured that simulated APs were suprathreshold in the nodes and subthreshold in the juxtaparanode and internode regions, indicating saltatory conduction. Of the 1600 simulated models, 138 met these criteria; for the present study, we randomly selected 50 models to comprise the young, control model cohort. Along most dimensions, the chosen cohort was approximately normally distributed (*Figure 2—figure supplement 1*). The g-ratio (ratio of axon to fiber diameter) among models in the cohort was 0.71±0.02, with total axon lengths of 1.2±0.1 cm.

## Simulating myelin alterations

We assumed that the effect of all myelin dystrophies observed empirically (e.g. *Figure 1*) could be modeled by removing lamellae from myelinated segments (demyelination, which reduces electrical insulation), and that remyelination could be modeled by replacing myelinated segments with two

shorter and thinner segments with a node in between. Evidence suggests that aging affects oligodendrocytes in several ways, including the ability for oligodendrocyte precursor cells to mature (*Dimovasili et al., 2023*). Knowing that individual oligodendrocytes myelinate axons of many different neurons, but without data quantifying how oligodendrocyte dystrophy affects myelination in individual axons, we assumed that myelin alterations were randomly distributed. To simulate demyelination, we varied two independent factors: the percentage of myelinated segments selected for demyelination along an axon (demyelination percentage), and the percentage of myelin lamellae removed from those segments (lamellae removal percentage). For each demyelination percentage (10%, 25%, 50%, and 75%; *Figure 3*), we generated 30 randomized lists of segments to demyelinate. Then for each of the 30 trials, each lamellae removal percentage was applied (25%, 50%, 55%, 60%, 65%, 70%, 75%, and 100%) to the chosen segments, for all 50 models in the control cohort. We then simulated the +380 pA current step, calculating the CV and the number of APs that propagated to the distal end of the axon. For each perturbation, we defined the CV change as the percentage change in CV induced relative to the CV of the corresponding unperturbed model. We also computed, for each perturbation, the percentage of AP failures at the distal axon end, relative to the number of APs observed at the first node.

To simulate remyelination three factors were varied: the percentage of myelinated segments initially demyelinated (demyelination percentage: 25%, 50%, 75%); the percentage of those affected segments which were then remyelinated (remyelination percentage: 25%, 50%, 75%, 100%); and the percentage of lamellae restored with remyelination (lamellae restoration percentage: 10%, 25%, 50%, 75%). We performed these remyelination simulations under two demyelination conditions: where the initially demyelinated segments had lost all their lamellae ('complete demyelination'), or had lost half their lamellae ('partial demyelination'). Sample remyelination perturbations shown in *Figure 4*. Remyelination was performed by replacing an affected (previously demyelinated) segment with two shorter segments, each including paranodes, juxtaparanodes, and an internode, and a new node between them that was identical to existing nodes. The number of lamellae on the shorter segments was determined by the lamellae restoration percentage. As before, for each demyelination percentage (25%, 50%, and 75%), we generated 30 randomized lists of segments to demyelinate. Segments to remyelinate were decided based on remyelination percentage. For example, in the case of 50% remyelination, every alternate (previously) demyelinated segment was remyelinated. Then for each trial, we applied each combination of remyelination percentage and lamellae restoration percentage to the chosen segments, for all models in the cohort, and simulated the current clamp protocol. For each remyelination condition, in addition to computing the percentage of AP failures, we defined CV recovery as the percentage improvement in CV relative to the CV change for the completely demyelinated case. For example, if a control model had a CV of 1 m/s, and a CV of 0.6 m/s after complete demyelination, then the CV change was –0.4/1.0 = –40%. If a subsequent remyelination condition led to a CV of 0.8 m/s (an increase of 0.2 m/s over the demyelinated case), the CV recovery was 0.2/0.4=50%.

## Statistical assessment of parameter importance

To identify which parameters of the multicompartment model had the greatest influence on axonal responses to demyelination and remyelination perturbations, we used least absolute shrinkage and selection operator (Lasso) regression (*Tibshirani, 1996*; *James et al., 2021*) implemented in MATLAB R2022a (Mathworks, Natick, MA, USA). Lasso fits a regression model to response variable, given predictor variables (observations of predictors) by selecting coefficients that minimize the quantity

$$\sum_{i=1}^{n} \left( y_i - \beta_0 - \sum_{j=1}^{p} \beta_j x_{ij} \right)^2 + \lambda \sum_{j=1}^{p} |\beta_j|,$$

for a given value of *James et al., 2021*. The former quantity is the residual sum of squares of the model. The latter quantity is the absolute sum of coefficients; including it in the minimization shrinks some coefficients to zero when $\lambda$ is sufficiently large, enabling Lasso to perform feature selection. The tuning parameter was selected to minimize the 10-fold cross-validation error; then for the selected value of $\lambda$, the regression was repeated using all available data. A total of 12 parameters served as predictor variables for the Lasso regression. We started with 6 of the 8 parameters used to construct

the cohort (axon diameter, node length, myelin length, and scale factors for leak, NaF, and KDR conductances), and combined two other parameters (number of myelin lamellae and lamella thickness) into one quantity: myelin thickness (their product). Past studies (*Goldman and Albus, 1968*; *Koles and Rasminsky, 1972*; *Moore et al., 1978*; *Gow and Devaux, 2008*) identified several other parameters that affect axonal propagation, including the g-ratio, axoplasmic resistance, axon capacitance, myelin resistance, myelin capacitance, and tight junction resistance. We added five of these parameters to the seven cohort parameters as predictor variables, omitting only g-ratio due to its high correlation with axon diameter and myelin thickness. We created response variables summarizing the effects of demyelination and remyelination on CV in each of the 50 members of the model cohort. AP failures were recorded as CV change of –100%. We did not include AP failure rates as a response variable, since CV changes precede AP failures. For demyelination, we averaged the CV change for all randomized trials in which 100% of lamellae were removed from 25%, 50%, and 75% of segments. For remyelination, we averaged the CV recovery in all randomized trials in which 25%, 50%, or 75% of segments were completely demyelinated, and then all affected segments were remyelinated as two shorter segments with 75% of lamellae added back. To facilitate comparison, we z-scored all predictor and response variables before performing Lasso separately on each response. We tested the predictive ability of the Lasso models by randomly selecting another 50 of the 138 models from the original hypercube that met the inclusion criteria, then simulating the specific myelin alteration protocols that

**Table 2.** Network model parameters.

| Parameter | Value |
|---|---|
| $g_{Eea}$ | $533.3/\sqrt{K_E}$ mV · ms |
| $g_{Een}$ | $490.64/\sqrt{K_E}$ mV · ms |
| $g_{Eia}$ | $67.2/\sqrt{K_E}$ mV · ms |
| $g_{Ein}$ | $7.4/\sqrt{K_E}$ mV · ms |
| $g_{IE}$ | $-138.6/\sqrt{K_I}$ mV · ms |
| $g_{II}$ | $-90.6/\sqrt{K_I}$ mV · ms |
| $\tau_E$ | 20 ms |
| $\tau_I$ | 10 ms |
| $\tau_a$ | 3 ms |
| $\tau_n$ | 50 ms |
| $\tau_g$ | 4 ms |
| $\tau_d$ | 200 ms |
| $\tau_f$ | 450 ms |
| $U$ | 0.03 |
| $V_T$ | 20 mV |
| $V_R$ | $-3.33$ mV |
| $\sigma_{EE}$ | 30° |
| $\sigma_{EI}$ | 35° |
| $\sigma_{IE}$ | 30° |
| $\sigma_{II}$ | 30° |
| $I_E^{ext}$ | $1.66\ \sqrt{K_E}$ mV |
| $I_I^{ext}$ | $1.5355\ \sqrt{K_E}$ mV |
| $I_{max,E}$ | 0.24 mV |
| $\varepsilon_E$ | 61.2° |

comprised the demyelination and remyelination response variables. We computed Pearson's correlation coefficient between the z-scored predicted vs. observed responses.

## Spiking neural network model

We adapted a neural network model (**Hansel and Mato, 2013**) to simulate the neural circuit in the dlPFC underlying spatial working memory during the oculomotor DRT (**Figure 5A**). The network model is composed of $N = 20,000$ leaky integrate-and-fire neurons, $N_E = 16,000$ excitatory (E) neurons (80%) and $N_I = 4000$ inhibitory (I) neurons (20%) with sparse probabilistic connections among all neuronal populations. A full description of this probabilistic version of the ring model can be found in **Hansel and Mato, 2013**. Here, we describe the essential features of the network and summarize all the parameter values in **Table 2**. The range of the interactions is represented by the parameters $\sigma_{EE}$, $\sigma_{EI}$, $\sigma_{IE}$, and $\sigma_{II}$. Each neuron receives $K = 500$ total inputs, where $K_E = 0.8\,K$ are excitatory inputs and $K_I = 0.2\,K$ are inhibitory inputs. The subthreshold membrane potential of each excitatory and inhibitory neuron ($i$) in the network is described by

$$\tau_E \frac{dV_{E,i}}{dt} = -V_{E,i} + I_{E,i}^{rec}\left(t\right) + I_E^{ext} + I_{E,i}^{cue}\left(t\right),$$

$$\tau_I \frac{dV_{I,i}}{dt} = -V_{I,i} + I_{I,i}^{rec}\left(t\right) + I_I^{ext},$$

where $\tau_E$ and $\tau_I$ are the membrane time constants for the E and I neurons, respectively. $I_{E/I,i}^{rec}(t)$ is the total recurrent synaptic current that each neuron receives from all the other neurons in the network connected to it. $I_{E/I}^{ext}$ is a constant background input, representing internal brain currents that come from outside the network. $I_{E,i}^{cue}(t)$ represents the transient sensory input to each E neuron, associated to a given direction of the stimulus, and only active during the cue period of the task. An AP is fired each time that the membrane potential of a neuron reaches the threshold value, $V_T$. The voltage of the membrane is reset to the baseline value, $V_R$, immediately after.

The total recurrent synaptic current for each neuron, $I_{E/I,i}^{rec}(t)$, is given by

$$I_{E/I,i}^{rec}\left(t\right) = I_{E/I,i}^{a}\left(t\right) + I_{E/I,i}^{n}\left(t\right) + I_{E/I,i}^{g}\left(t\right),$$

where subindices $a$ and $n$ represent the excitatory AMPA and NMDA glutamatergic receptors, and $g$ an inhibitory GABA receptor. Each synaptic current is given as in **Hansel and Mato, 2013**, with synaptic decay time constants for each receptor, $\tau_a, \tau_n$, and $\tau_g$, respectively. Short-term plasticity was incorporated to the excitatory-to-excitatory synaptic connections through the variables $u$ and $x$, given by **Markram et al., 1998**

$$\tau_f \frac{du_i}{dt} = U - u_I$$

$$\tau_d \frac{dx_i}{dt} = \left(1 - x_I\right).$$

$x$ and $u$ are updated as $x \rightarrow x\left(1 - u\right)$ and $u \rightarrow u + U * \left(1 - u\right)$, each time that there is a presynaptic spike. The variable $x$ represents the amount of available neurotransmitter resources in the presynaptic terminal and $u$ is the utilization parameter, indicating the residual calcium level (**Bertram et al., 1996**; **Zucker and Regehr, 2002**; **Mongillo et al., 2008**). With each spike, amount $ux$ of the available resources is used to produce the postsynaptic current. Thus, $x$ is reduced, representing neurotransmitter depletion, and $u$ is increased, representing the calcium influx into the presynaptic terminal and its effect on release probability. Between spikes, $x$ and $u$ recover to their baseline levels ($x = 1$ and $u = U; 0 < x < 1$) with time constants $\tau_d$ and $\tau_f$. We set $\tau_f > \tau_d$ so that they facilitate signal transmission (**Tsodyks et al., 1998**).

The sensory input current to excitatory neuron $i$ during the cue period, and for a specific location of the stimulus, $I_{E,i}^{cue}(t)$, is given by

$$I_{E,i}^{cue}\left(t\right) = I_{max,E}\,e^{-\frac{1}{2}\left(\frac{\theta_i - \theta_{cue}}{\varepsilon_E}\right)^2},$$

where $\theta_i$ is the preferred direction of the neuron $i$, given by its position on the ring, $\theta_{cue}$ is the cue direction, and $I_{max,E}$ and $\varepsilon_E$ are the amplitude and the width of the sensory input current, respectively.

## Network simulations with spatially modulated correlations

To introduce spatially modulated correlations in the model (*Figure 5—figure supplement 1*), we reduced the strength of the constant background inputs to excitatory and inhibitory neurons, $I_E^{ext}$ and $I_I^{ext}$, by a factor of 0.5 and provide additional external input from a population of Poisson neurons. The parameters were chosen such that the time-averaged total input (the sum of the reduced constant input and the inputs from the Poisson population) is the same as in our default network without Poisson inputs. The external population is composed of $N^{ext}$ = 16,000 Poisson neurons that fire with a constant firing rate of $r^{ext}$ = 9.72 Hz. These neurons are connected through AMPA synapses to excitatory neurons in the network with a strength of $g_{Ea}^{ext} = 0.5\, g_{Eia}$ and to inhibitory neurons with a strength of $g_{Ia}^{ext} = 0.4625\, g_{Eia}$. The average total number of synaptic inputs from Poisson neurons that a neuron receives is $K^{ext}$ = 1000. Crucially, the connections from the Poisson population to neurons in the network are spatially structured, with a Gaussian connection profile similar to the recurrent connections in the network (ring structure) and with the interaction range determined by $\sigma^{ext}$ = 20°. Thus, neurons in the network receive shared inputs with a spatial structure and this leads to spatial correlations in the network neurons (*Rosenbaum et al., 2017*).

## Simulating a cohort of young (control) networks

Using the Brian2 simulator (based on Python), we simulated 10 different networks, all with the same parameters, but each with a different connectivity profile. For each network, we ran 280 trials, with different initial conditions and cue positions. All networks performed the DRT – with 2 s fixation period, 1 s cue period, and 4 s delay period – and maintained the memory of the stimulus during the whole delay period of the task (*Figure 5B, i* and *Figure 5—figure supplement 1A*, left panel).

## Modeling the effects of demyelination and remyelination in the network model

### Modeling AP propagation failures in the network

The network model is composed of point neurons without an explicit model of the axon. To effectively model the AP failures at the distal end of the axons quantified with the single neuron model under the different demyelination and remyelination conditions, the AP failure rate was adjusted to the values produced by the single neuron model. To do this, we perturbed the 10 control networks by designing a probabilistic model of spike transmission from the excitatory presynaptic neurons to both the excitatory and inhibitory postsynaptic neurons. From the single neuron model, for each demyelination/remyelination condition, we quantified the probability of AP failure for each of the neurons in the control cohort, as well as the percentage of those neurons that shared the same probabilities of failure. That is, the percentage of neurons that had probability of failure = 0, probability of failure = 1 or any other probability. Then, we computed the probability of transmission, $p_{transmission} = 1 - p_{failure}$, and we specified $p_{transmission}$ for the corresponding percentages of excitatory neurons in the networks. Thus, in the network model, we took into account the heterogeneity observed in the single neuron model under each demyelination/remyelination condition.

### Modeling conduction velocity slowing in the network

To explore the effect of CV slowing along the axons of model neurons, we simulated 20 young, control networks and 20 perturbed networks with AP failure rates adjusted for the case of single model neurons with 50% of the segments demyelinated along the axons by removing 60% of the myelin lamellae (we ran 280 trials for each network). Then, we added random delays uniformly distributed with a minimum value of 0 ms in both cases, a maximum value of 100 ms in the control networks, and a maximum values of 40 and 85 ms in the perturbed networks, in both the AMPA and NMDA excitatory connections to both *E* and *I* neurons (*Figure 5—figure supplement 2*). These large values were chosen because we wanted to illustrate the potential effect of CV slowing in our network and smaller, more realistic, values did not have any effect.

## Quantification of normal myelin sheaths in groups of model neurons containing both intact and demyelinated axons

We created different groups of 50 total model neurons containing, among those 50, different random amounts of neurons with intact axons and with demyelinated axons, with either 10%, 25%, 50%, or 75% of segments demyelinated. In each group, the distribution of the demyelinated segments along the altered axons was also randomly chosen among the 30 possible distributions simulated with the single neuron model. We sorted the axons in each group by locating their origin aligned in the same position and, up to the maximum length of the shortest axon, we divided the axons longitudinally in sections of 0.5 μm, and calculated the percentage of normal, unperturbed myelin sheaths in each section (*Figure 7A*). Then, we averaged across all sections and we picked 60 groups that had over 80% of normal myelin sheaths. According to the proportion of intact and demyelinated axons (with either 10%, 25%, 50%, or 75% of segments affected) in each group, we adjusted the distribution of AP transmission probability ($p_{transmission}$; $p_{transmission} = 1$ for intact axons) in 1 of the 10 control networks. To do this we considered that the perturbed axons in 40 of the groups were completely demyelinated (all lamellae removed) and that in the remaining 20 groups, they had 75% of lamellae removed.

## Quantification of new myelin sheaths in groups of model neurons containing both intact and remyelinated axons

We again created different groups of 50 total model neurons containing different random amounts of neurons with intact and remyelinated axons. The remyelinated axons had either 25%, 50%, or 75% of segments remyelinated (with two shorter and thinner myelin sheaths), following previous demyelination of either 25%, 50%, or 75% of the segments. In each group, the distribution of the remyelinated segments along the altered axons was randomly chosen among the 30 possible distributions. As before, we sorted the axons in each group by locating their origin aligned in the same position, we divided the axons longitudinally in sections of 0.5 μm (up to the maximum length of the shortest axon), and calculated the percentage of remyelinated segments in each section (*Figure 8A*). Then, we averaged across all sections, we multiplied by two to calculate the number of new myelin sheaths, and we picked 22 groups that had below 45% of new myelin sheaths. According to the proportion of intact and remyelinated axons (with different percentages of remyelinated segments) in each group, we adjusted the distribution of $p_{transmission}$ in the same control network as before for the case of remyelination following partial demyelination by adding 25% of the lamellae back.

## Measures of working memory performance

The remembered location was obtained using a population vector decoder and network performance was quantified by the following four measures that describe the quality of the activity bump maintaining a memory of the stimulus during the delay period: the memory strength, the memory duration, the drift rate, and the diffusion constant.

### Memory strength

The memory strength was defined as the modulus $M(t)$ of the population vector, $Z(t)$, which characterizes the spatial modulation of the excitatory neuronal activity at time $t$, given by

$$Z(t) = \frac{\sum_j r_j(t) e^{i\theta_j}}{\sum_j r_j(t)} = M(t) e^{i\Psi(t)}.$$

$r_j(t)$ is the firing rate of neuron $j$ with preferred direction $\theta_j$. Firing rates $r_j(t)$ were estimated as spike counts in a 250 ms window. A memory strength $M(t)$ close to 0 indicates homogeneous activity of the network and $M(t)$ close to 1 indicates a sharply modulated activity (*Figure 5C*).

### Decoded cue location

$\Psi(t)$ is the argument of $Z(t)$ and indicates the remembered stimulus location. That is, the center of the activity bump (*Figure 5B*).

## Memory duration

The memory duration was defined as the time from the delay onset until the time where the memory strength decayed below a fixed threshold that we set at 0.4. That is, the memory duration is the interval during which the bump of neural activity is reliably maintained (*Figure 5C*).

## Memory drift

The drift was defined as the bias of the estimator, which is given by

$$b_{est}(t) = \langle \Phi(t) \rangle - \theta_{cue}.$$

That is, the difference of the trial average of the estimates $\Phi(t)$ and the true value $\theta_{cue}$ (*Esnaola-Acebes et al., 2022*; *Figure 5E*). The drift rate is the slope of a linear fit of the drift during the memory duration time (*Figure 6—figure supplement 2*).

## Memory diffusion

The diffusion was defined as the variance of the estimator, given by

$$\sigma_{est}^2(t) = \left\langle (\Phi(t) - \langle \Phi(t) \rangle)^2 \right\rangle,$$

the variation of the estimates about their mean value (*Esnaola-Acebes et al., 2022*; *Figure 5D*). The diffusion rate is the slope of a linear fit of the diffusion during the memory duration time (*Figure 6B*).

## Acknowledgements

We acknowledge the use of Fenix Infrastructure resources, which are partially funded from the European Union's Horizon 2020 research and innovation program through the ICEI project under the grant agreement No. 800858, and the research cluster at Franklin and Marshall College, funded through NSF grant 1925192. We thank David Latimer and Albert Compte for sharing computer code, as well as Jason Brooks and Tony Weaver for technical assistance. We thank the CERCA Programme/Generalitat de Catalunya for institutional support. This work was supported by NIH/NIA grant R01 AG059028, NIH 1R01AG071230-01, and grant PCI2020-112035 from MCIN/AEI/10.13039/501100011033 and the European Union 'NextGenerationEU'/PRTR. This work was supported by the Spanish State Research Agency, through the Severo Ochoa and María de Maeztu Program for Centers and Units of Excellence in R&D (CEX2020-001084-M). This work utilized the research cluster at Franklin and Marshall College which was funded through NSF grant 1925192.

## Additional information

### Funding

| Funder | Grant reference number | Author |
| --- | --- | --- |
| National Institute on Aging | R01 AG059028 | Jennifer I Luebke |
| MCIN/ AEI/10.13039/501100011033 and the European Union "NextGenerationEU"/PRTR | PCI2020-112035 | Sara Ibañez Klaus Wimmer |
| Spanish State Research Agency, through the Severo Ochoa and María de Maeztu Program for Centers and Units of Excellence in R&D | CEX2020-001084-M | Sara Ibañez Klaus Wimmer |
| National Science Foundation | 1925192 | Christina M Weaver |

| Funder | Grant reference number | Author |
|---|---|---|
| Horizon 2020 - Research and Innovation Framework Programme | ICEI project grant agreement No. 800858 | Sara Ibañez Klaus Wimmer |
| National Institute on Aging | R01 AG071230 | Jennifer I Luebke |

The funders had no role in study design, data collection and interpretation, or the decision to submit the work for publication.

## Author contributions

Sara Ibañez, Conceptualization, Resources, Data curation, Software, Formal analysis, Funding acquisition, Validation, Investigation, Visualization, Methodology, Writing – original draft, Writing – review and editing; Nilapratim Sengupta, Conceptualization, Data curation, Software, Formal analysis, Validation, Investigation, Visualization, Methodology, Writing – original draft, Writing – review and editing; Jennifer I Luebke, Conceptualization, Resources, Data curation, Supervision, Funding acquisition, Investigation, Writing – original draft, Project administration, Writing – review and editing; Klaus Wimmer, Christina M Weaver, Conceptualization, Resources, Data curation, Software, Formal analysis, Supervision, Funding acquisition, Validation, Investigation, Visualization, Methodology, Writing – original draft, Project administration, Writing – review and editing

## Author ORCIDs

Sara Ibañez ⓘ https://orcid.org/0000-0001-7563-6968
Nilapratim Sengupta ⓘ https://orcid.org/0000-0003-1024-559X
Jennifer I Luebke ⓘ http://orcid.org/0000-0003-1399-6073
Klaus Wimmer ⓘ https://orcid.org/0000-0003-2973-3462
Christina M Weaver ⓘ https://orcid.org/0000-0002-7744-4289

Reviewer #1 (Public review): https://doi.org/10.7554/eLife.90964.3.sa1
Reviewer #2 (Public review): https://doi.org/10.7554/eLife.90964.3.sa2
Author response https://doi.org/10.7554/eLife.90964.3.sa3

## Additional files

### Supplementary files
• MDAR checklist

### Data availability

The single neuron and network models for this study are available on ModelDB (https://modeldb.science), accession number 2014821.

The following dataset was generated:

| Author(s) | Year | Dataset title | Dataset URL | Database and Identifier |
|---|---|---|---|---|
| Ibañez S, Sengupta N, Luebke JI, Wimmer K, Weaver CM | 2024 | Myelin dystrophy impairs signal transmission and working memory decline in a multiscale model of the aging prefrontal cortex | https://modeldb.science/2014821 | ModelDB, 2014821 |

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
