## [Editor Report · eLife assessment]

This manuscript reports a **valuable** computational study of the effects of axon de-myelination and re-myelination on action potential speed and propagation failure. The manuscript presents **solid** evidence for the effects of de- and re-myelination in different models of working memory, with potential implications in disorders such as multiple sclerosis. The exposition of the manuscript is targeted for researchers interested in biophysical models of cognitive deficits.

---

## [Referee Report · Reviewer #1 (Public review)]

Summary:

The authors study the effects of myelin alterations in working memory via the complementary use of two computational approaches: one based on the de- and re-myelination in multicompartmental models of pyramidal neurons, and one based on synaptic changes in a spiking bump attractor model for spatial working memory. The first model provides the most precise angle (biophysically speaking) of the different effects (loss of myelin lamella or segments, remyelination with thinner and shorter nodes, etc), while the second model allows to infer the consequences of myelin alterations in working memory performance, including memory stability, duration, and bump diffusion, while also exploring the case of myeling alterations in a novel silent working memory model. The results indicate (i) a slowing down and failure of propagation of spikes with demyelination and partial recovery with remyelination, with detailed predictions on the role of nodes and myelina lamella, and (ii) a decrease in memory duration and an increase in memory drift as a function of the demyelination, in agreement with multiple experimental studies.

Strengths:

Overall, the work offers a very interesting approach of a topic which is hard to accomplish experimentally --therefore the computational take is entirely justified and extremely useful. The authors carefully designed the computational experiments to shed light into the demyelination effects on working memory from multiple levels of description, increasing the reliability of their conclusions. I think this work provides now convincing evidence and has the potential to be influential in future studies of myelin alterations (and related disorders such as multiple sclerosis).

Weaknesses:

In its current form, the authors have improved the clarity of the results and the model details, and have provided a new set of simulations to complement and reinforce the original ones (including the development of a new spatial working memory model based on silent working memory principles). I do not appreciate any significant weaknesses at this point.

---

## [Referee Report · Reviewer #2 (Public review)]

This paper analyzes the effect of axon de-myelination and re-myelination on action potential speed, and propagation failure. Next, the findings are then incorporated in a standard spiking ring attractor model of working memory.

I think the results are not very surprising or solid and there are issues with method and presentation.

The authors did many simulations with random parameters, then averaged the result, and found for instance that the Conduction Velocity drops in demyelination. It gives the reader little insight into what is really going on. My personal preference is for a well understood simple model rather than a poorly understood complex model. The link between the model outcome of WM and data remains qualitative and is further weakened by the existence of known other age-related effects in PFC circuits.

Comments on revised version:

The paper has improved in the revision, although I still think a reduced model would have been nice.

---

## [Author Response]

The following is the authors’ response to the current reviews.

We thank the reviewers for their overall careful evaluation of our work, the constructive criticism, and their many helpful suggestions. We feel that our revision built on the strengths identified by the reviewers, and addressed all the concerns they have raised. Both reviewers recognize that our revisions have improved the paper. Since the first submission we have:

Rewritten large parts of the papers to improve clarity and make it more concise where possibleSimulated an alternative working memory model, as recommended by Reviewer 1Included 4 new/revised supplementary figures, following the reviewer’s suggestions for additional analysis.

Below we provide a brief response to the Reviewers’ comments on our manuscript revision.

**Reviewer #1: Public Review:**
Strengths:Overall, the work offers a very interesting approach of a topic which is hard to accomplish experimentally --therefore the computational take is entirely justified and extremely useful. The authors carefully designed the computational experiments to shed light into the demyelination effects on working memory from multiple levels of description, increasing the reliability of their conclusions. I think this work provides now convincing evidence and has the potential to be influential in future studies of myelin alterations (and related disorders such as multiple sclerosis).Weaknesses:In its current form, the authors have improved the clarity of the results and the model details, and have provided a new set of simulations to complement and reinforce the original ones (including the development of a new spatial working memory model based on silent working memory principles). I do not appreciate any significant weaknesses at this point.

We thank the reviewer for these positive comments on our revision and for the suggestion of adding the silent memory model, as we feel this has strengthened our findings.

**Reviewer #2: Public Review:**
This paper analyzes the effect of axon de-myelination and re-myelination on action potential speed, and propagation failure. Next, the findings are then incorporated in a standard spiking ring attractor model of working memory.I think the results are not very surprising or solid and there are issues with method and presentation.The authors did many simulations with random parameters, then averaged the result, and found for instance that the Conduction Velocity drops in demyelination. It gives the reader little insight into what is really going on. My personal preference is for a well understood simple model rather than a poorly understood complex model. The link between the model outcome of WM and data remains qualitative and is further weakened by the existence of known other age-related effects in PFC circuits.Comments on revised version:The paper has improved in the revision, although I still think a reduced model would have been nice.

As noted above, in addition to our spiking bump attractor model, our revision includes a second network-level model: an activity-silent working memory model for continuous features. We found qualitatively similar effects as in our bump attractor network model, showing that our main conclusions do not critically depend on the exact working memory mechanism (active vs. activity-silent). This new model was described in two new supplementary figures and a new paragraph in the Results section.

We did not add a reduced model in our revision to this paper, since neither reviewer explicitly recommended that we add one. As we noted in our private response to reviewers that accompanied our revision: we share the view that understanding simple models can provide critical insights into brain function (and we believe that many of our papers related to attractor dynamics in working memory and decision-making fall into this category, e.g. Wimmer et al. 2014, Esnaola-Acebes et al. 2022, Ibañez et al 2020). We disagree with the reviewer on an important point: we feel that the model complexity that we have chosen is appropriate and necessary to study the phenomenon at hand. Our modeling efforts are principled, with complexity added as necessary. We started with a biophysical single neuron model with firing dynamics fit to empirical data in pyramidal neurons of rhesus monkey dlPFC (Rumbell et al. 2016) – the same type of neurons and cortical region analyzed in the Peters et al. work on structural changes to myelin seen during aging (e.g., Figure 1). Because simple models do not accurately capture the CV along thin axons like those in the PFC, we attached a multicompartment axon with detailed myelinated segments, and constructed a cohort of feasible models. We then used this cohort to get quantitative estimates of the effects of variable degrees of demyelination and remyelination. This would not be possible with a simpler model. We then study the consequences of de- and re-myelination in a spiking neural network model. Again, we could not use a simpler model (e.g. a firing rate attractor model) without making gross assumptions about how demyelination affects circuit function. In sum, we believe that our models are relatively simple but comprehensive given the phenomenon that we are studying.

The reviewer is correct in that there exist “known other age-related effects in PFC circuits”. These are reviewed in the introduction and we discuss future extensions of our model that would incorporate those effects as well. It is important to note that this is the first comprehensive study of demyelination effects in aging PFC, demonstrating that myelin changes alone predict working memory changes associated with aging.

While we agree that averaging results about different parameter sets provide a limited understanding of the system, we persist in our belief that such analyses provide an important baseline. We acknowledge that results vary across our model cohort; this is why we included the heatmaps of our single cell model perturbation results (Figure 3 and Supplementary Figure 3), and simulated network models representing a heterogeneity of neuronal axons with healthy and altered myelin sheaths in different degrees, as likely occurs in the aging brain (Figures 7 and 8). The model framework we present here is well-suited for more targeted analyses and better insights, including those which we are pursuing currently.

The following is the authors’ response to the original reviews.

We thank the reviewers for their careful evaluation of our work, the constructive criticism, and their many helpful suggestions. We feel that our revision builds on the strengths identified by the reviewers, and addresses all the concerns they have raised. We have:

Rewritten large parts of the papers to improve clarity and make it more concise where possibleSimulated an alternative working memory modelIncluded 4 new/revised supplementary figures, following the reviewer’s suggestions for additional analysis

**Reviewer #1 (Public Review):**
Summary:The authors study the effects of myelin alterations in working memory via the complementary use of two computational approaches: one based on the de- and re-myelination in multicompartmental models of pyramidal neurons, and one based on synaptic changes in a spiking bump attractor model for spatial working memory. The first model provides the most precise angle (biophysically speaking) of the different effects (loss of myelin lamella or segments, remyelination with thinner and shorter nodes, etc), while the second model allows to infer the consequences of myelin alterations in working memory performance, including memory stability, duration, and bump diffusion. The results indicate (i) a slowing down and failure of propagation of spikes with demyelination and partial recovery with remyelination, with detailed predictions on the role of nodes and myelina lamella, and (ii) a decrease in memory duration and an increase in memory drift as a function of the demyelination, in agreement with multiple experimental studies.Strengths:Overall, the work offers a very interesting approach of a topic which is hard to accomplish experimentally --therefore the computational take is entirely justified and extremely useful. The authors carefully designed the computational experiments to shed light into the demyelination effects on working memory from multiple levels of description, increasing the reliability of their conclusions. I think this work is solid and has the potential to be influential in future studies of myelin alterations (and related disorders such as multiple sclerosis).

We thank the reviewer for these positive comments on our manuscript.

Weaknesses:In its current form, the study still presents several issues which prevent it from achieving a higher potential impact. These can be summarized in two main items. First, the manuscript is missing some important details about how demyelination and remyelination are incorporated in both models (and what is the connection between both implementations). For example, it is unclear whether an unperturbed axon and a fully remyelinated axon would be mathematically equivalent in the multicompartment model, or how the changes in the number of nodes, myelin lamella, etc, are implemented in the spiking neural network model.

We thank the reviewer for these suggestions to improve the clarity of our manuscript. A ‘fully remyelinated’ axon is not mathematically equivalent to the unperturbed axon: it has shorter and thinner myelinated segments, and additional nodes in between. This is consistent with empirical observations in rhesus monkey dlPFC, as reviewed in Peters et al. (2009): a 90% increase in paranode profiles, and myelin sheaths that were thinner than expected for the size of the enclosed axon. With no empirical observations of fewer numbers of nodes (but rather, the opposite) or bare sections of axon, we assumed that the remyelination process also creates new nodes (which are identical to existing nodes), as also modeled in Scurfield & Latimer (2018). We have added two new sentences to the results to clarify this fact, before presenting the first set of results for the single cell model: (starting at line 137):

“To simulate demyelination, we removed lamellae from selected myelinated segments; for remyelination we replaced a fraction of myelinated segments by two shorter and thinner segments with a node in between. As such, a ‘fully remyelinated axon’ had all the demyelinated segments subsequently remyelinated, but with fewer lamellae and additional nodes compared to the unperturbed control case, consistent with empirical observations (Peters,
2009).”

We also state the maximal amount of remyelination more explicitly in the Results, starting on lines 164-165: "We next examined the extent to which remyelination with shorter and thinner segments, occurring after demyelination, restored axonal AP propagation (Figure 4).”

Also on line 192-193: “Remyelinating all affected segments with 75% of lamellae (the maximal amount of remyelination) nearly eliminated AP failures (1.8 ± 1.1%).”

Finally, in Methods we also clarified the structure of the added node (starting at line 634): “Remyelination was performed by replacing an affected (previously demyelinated) segment with two shorter segments, each including paranodes, juxtaparanodes, and an internode, and a new node between them that was identical to existing nodes.”

We have also provided further details describing how myelin dystrophy was simulated in the network model in Results (lines 243 - 249) and in Methods (lines 722 - 747). How myelin alterations have been implemented in the network model is one of the questions of the reviewer (Question 5 in Reviewer #1: Recommendations for the Authors_)._ We have addressed this question by describing in detail how we adjusted CV and AP failure rate to the values produced by the multicompartment neuron model. Please see our answer to Question 5 for the details.

Second, it is unclear whether some of the conclusions are strong computational predictions or just a consequence of the model chosen. For example, the lack of effect of decreasing the conduction velocity on working memory performance could be due to the choice of considering a certain type of working memory model (continuous attractor), and therefore be absent under other valid assumptions (i.e. a silent working memory model, which has a higher dependence on temporal synaptic dynamics).

Whether some conclusions are strong predictions or just a consequence of the model chosen is an important concern and indeed a general problem of computational modeling of working memory. For example, Stein et al. (Stein et al. Towards biologically constrained attractor models of schizophrenia, *Curr. Opin. Neurobiol. 2021)* showed that opposed manipulations of E/I ratio can produce the same behavioral pattern in different alternative, plausible biological network models. As long as we do not fully understand the neural mechanisms underlying working memory, modeling studies of how alterations (e.g. in E/I ratio or in the reliability and timing of axonal transmission, as we did here) affect circuit function need to be interpreted critically and tested against new experimental data.

One way to strengthen model predictions is by showing that different computational models make similar predictions. To do this, we implemented an activity-silent working memory model for continuous features, as suggested by the reviewer, and we found qualitatively similar effects as in our bump attractor network model. Thus, our main conclusions do not critically depend on the exact working memory mechanism (active vs. activity-silent).

In the revised manuscript, we have added two new supplementary figures (Supplementary Figure 8 and 9, see the next page) and a new paragraph in the Results section about activity silent working memory (starting at line 319):

“Alternative working memory mechanisms. Working memory in our neural network is maintained in an attractor state with persistent neural activity (Compte et al., 2000; Hansel and Mato, 2013). Other mechanisms have been proposed, including that working memory maintenance may rely on activity-silent memory traces (Mongillo et al., 2008; Stokes, 2015; Barbosa et al., 2020). In activity-silent models, a slowly decaying transient of synaptic efficacy preserves information without the need for persistent ongoing activity. We implemented an activity-silent model, to our knowledge the first one for continuous spatial locations, and tested how working memory performance is affected by AP failures and propagation delays. We found that AP failures corresponding to demyelination caused working memory errors qualitatively similar to the delay-active network (Supplementary Figure 8). On the other hand, increasing propagation delays did not lead to additional working memory errors, unless we include unrealistically high values (uniform distribution in the range of 0 to 100 ms; Supplementary Figure 9). These results are qualitatively similar to the delay active network model. Thus, our main findings do not critically depend on the exact working memory mechanism (active vs. activity-silent).”

**Author response image 1. sa3fig1:** Action potential failures impair working memory performance in a network model with activity-silent memory traces. (A) Spiking and synaptic activity in an unperturbed, activity-silent working memory model. Top: Raster plot showing the activity for each excitatory neuron (labeled by its preferred direction) in a single trial with a cue stimulus presented at 180°. We modified our spiking neural network model such that it does not show elevated persistent firing throughout the delay period (see Figure 5B for comparison). In particular, we reduced the external background input to excitatory neurons by a factor of 3.61% and we increased the cue stimulus amplitude by 12.5%. Even though spiking activity decays to baseline (close to 0 Hz), a memory trace is imprinted in enhanced synaptic strength due to short-term synaptic facilitation (Mongillo et al., 2008). Selective spiking activity is recovered by a non-selective constant input applied during 300 ms to all excitatory neurons during the two reactivation periods (marked by yellow and green rectangles in the raster plot). The amplitude of the input was 11 mV during the first and 13 mV during the second reactivation period. Reactivation periods are marked in light gray shading in the remaining panels below and the cue period is indicated by dark gray shading. Firing rates (second row), synaptic facilitation variable u (third row), and synaptic depression variable x (bottom row) for the same trial, averaged for 500 neurons around the neuron with 180° as preferred direction (solid lines) and around the neuron with 0° as preferred direction (dashed lines). Note that reactivation recovers the activity bump (C) but also causes elevated firing and subsequent enhancement of synapses at all positions in the networks. (B) Activity in a network with demyelination of 50% of the myelinated segments by removing 60% of the myelin lamellae. AP failures lead to reduced firing rates in the cue and early delay periods and consequently to weaker synaptic enhancement. (C) Average spike counts of the excitatory neurons during the cue period (black lines), and the two reactivation periods indicated in the raster plots in A and B (yellow and green lines). Solid lines correspond to the control network and dashed lines to the perturbed network. (D) Memory strength as a function of time for the control and perturbed networks. (E-F) Trajectories of the bump center (i.e., remembered cue location) read out from the neural activity across the cue and delay periods using a population vector (see Methods). Cue position was 180° in all trials. The perturbed network (F) shows larger working memory errors towards the end of the delay period compared to the control network (E).

**Author response image 2. sa3fig2:** Effect of propagation delays on control and perturbed activity-silent network models. (A) Memory strength during the whole simulation time for the young, control networks relying on activity-silent working memory (Supplementary Figure 8) with zero propagation delays (blue line), and with propagation delays from a uniform distribution with a range between 0 and 40 ms (yellow line) and between 0 and 100 ms (orange line). (B) Memory strength for perturbed networks when demyelinating 25% of the myelinated segments by removing 50% of the myelin lamellae, without delays (red line), and with uniformly distributed delays between 0 and 40 ms (light gray line) and between 0 and 100 ms (black line). The cue period is indicated by dark gray shading and reactivation periods are marked in light gray. Memory strength was calculated by averaging across 280 trials for one network. Shaded areas indicate SEM for each case. For the young, control networks (A), working memory was not affected by including delays of up to 40 ms. Unrealistically long delays ranging up to 100 ms did cause an impairment (the longest delays found for the most extreme perturbation condition – demyelination of 75% of the segments by removing 100% of the myelin lamellae – were of 49.9 ms on average). When also incorporating AP failures to the networks (B), we observed a similar trend. For this perturbation condition, delays of up to 40 ms were already much larger than the delays quantified in the single neuron model (for the case of 25% of the segments demyelinated by removing 50% of the myelin lamellae, the average delay in the cohort was 3.75 ms).

With additional simulations to address these issues, I consider that the present study would become a convincing milestone in the computational modeling of myelin-related models, and an important study in the field of working memory.

Again, we would like to thank the reviewer for the positive comments. We have addressed all the main issues raised (see below our response to the “recommendations for the authors”).

**Reviewer #2 (Public Review):**
This paper analyzes the effect of axon de-myelination and re-myelination on action potential speed, and propagation failure. Next, the findings are then incorporated in a standard spiking ring attractor model of working memory.I think the results are not very surprising or solid and there are issues with method and presentation.The authors did many simulations with random parameters, then averaged the result, and found for instance that the Conduction Velocity drops in demyelination. It gives the reader little insight into what is really going on. My personal preference is for a well understood simple model rather than a poorly understood complex model. The link between the model outcome of WM and data remains qualitative, and is further weakened by the existence of known other age-related effects in PFC circuits.

We thank the reviewer for the critical assessment of our work. We share the view that understanding simple models can provide critical insights into brain function (and we believe that many of our papers related to attractor dynamics in working memory and decision making fall into this category, e.g. Wimmer et al. 2014, Esnaola-Acebes et al. 2022, Ibañez et al 2020). However, we respectfully disagree with the reviewer on an important point: the model complexity that we have chosen is appropriate and necessary to study the phenomenon at hand. Our modeling efforts are principled, with complexity added as necessary. We started with a biophysical single neuron model with firing dynamics fit to empirical data in pyramidal neurons of rhesus monkey dlPFC (Rumbell et al. 2016) – the same type of neurons and cortical region analyzed in the Peters et al. work on structural changes to myelin seen during aging (e.g., Figure 1). Because simple models do not accurately capture the CV along thin axons like those in the PFC, we attached a multicompartment axon with detailed myelinated segments, and constructed a cohort of feasible models. We then used this cohort to get quantitative estimates of the effects of variable degrees of demyelination and remyelination. This would not be possible with a simpler model. We then study the consequences of de- and re-myelination in a spiking neural network model. Again, we could not use a simpler model (e.g. a firing rate attractor model) without making gross assumptions about how demyelination affects circuit function. In sum, we believe that our models are relatively simple but comprehensive given the phenomenon that we are studying.

The reviewer is correct in that there exist “known other age-related effects in PFC circuits”. These are reviewed in the introduction and we discuss future extensions of our model that would incorporate those effects as well. It is important to note that this is the first comprehensive study of demyelination effects in aging PFC, demonstrating that myelin changes alone predict working memory changes associated with aging.

The specific issues about modeling choices and interpretation of the results are discussed below.

Both for the de/re myelination the spatial patterns are fully random. Why is this justified?

We agree that myelin dystrophy during aging could be non-random, that is, localized to certain regions of an axon. Our collaborators (Drs Jennifer Luebke, Maya Medalla, and Patrick Hof) are currently addressing this question using 3D electron microscopy and immunohistochemistry on axons of individual neurons and their associated myelin, but results are not available yet. Early on in this study we examined how the location of myelin alterations affected AP propagation. Focusing demyelination along a section of axon led to more AP slowing and failure than when spatially randomized. Likewise, remyelination of such spatially localized dystrophy led to greater recovery, as there were fewer transitions between long and short internodes (Supplemental Figure 4). Since otherwise the effects in the localized cases were largely similar to those in the spatially random case (see Author response image 3 below), for brevity in this paper we assumed myelin alterations were randomly distributed. Our next paper, extending this study to collateralized axons and which was presented as a poster at the 2023 Society for Neuroscience meeting, will include an examination of localized myelin dystrophy.

**Author response image 3. sa3fig3:** Effect of localized myelin alterations on CV change. Myelin alterations were either focused on the third of myelinated segments closest to the initial segment (‘proximally clustered’), the third of myelinated segments furthest from the initial segment (‘distally clustered’), or distributed according to a uniform distribution as in the current study. For demyelination, all lamellae were removed from 25% of myelinated segments (showing mean +/- SEM of all 50 cohort models, 30 randomized trials each). For remyelination, affected segments were replaced by two shorter segments with 75% of the original lamellae thickness and a node in between.

We have added two sentences in Methods to justify this assumption more clearly (line 510): “Evidence suggests that aging affects oligodendrocytes in several ways, including the ability for oligodendrocyte precursor cells to mature (Dimovasili et al., 2022). Knowing that individual oligodendrocytes myelinate axons of many different neurons, but without data quantifying how oligodendrocyte dystrophy affects myelination in individual axons, we assumed that myelin alterations were randomly distributed.”

We have also added a sentence in the Discussion alluding to our upcoming study (line 434): “Our model can also be extended to explore interactions between spatially localized myelin perturbations (such as those seen in multiple sclerosis) and axon collateralization (Sengupta et al., 2023), which would affect the distance-dependence of AP failures.”

Similarly, to model the myelin parameters were drawn from uniform distributions, Table 1 (I guess). Again, why is this reasonable?

The reviewer is correct that our initial Latin hypercube sample generated a uniform distribution. However, parameters of the random sample of models selected as biologically feasible were not uniformly distributed. We have added a new figure (Supplementary Figure 1A) to illustrate the parameter distributions, and have added two sentences in Methods (starting on line 596):

“Of the 1600 simulated models, 138 met these criteria; for the present study, we randomly selected 50 models to comprise the young, control model cohort. Along most dimensions, the chosen cohort was approximately normally distributed (Supplementary Figure 1). The g-ratio (ratio of axon to fiber diameter) among models in the cohort was 0.71 ± 0.02, with total axon lengths of 1.2 ± 0.1 cm.”

**Author response image 4. sa3fig4:** Distribution of parameters and conduction velocities in the single neuron model cohort. (A) Histograms of axon morphology parameters of models selected for the single neuron cohort. Top: axon diameter: middle, length of unperturbed myelin segments; bottom: total myelin thickness in unperturbed segments, computed as the product of lamella thickness and number of lamellae. (B) Histograms of the CV for the 50 axons of the unperturbed model cohort (top), and representative demyelination and remyelination perturbations: mild demyelination (removing 25% of lamellae from 25% of the myelinated segments, second row); severe demyelination (removing all lamellae from 75% of the myelinated segments, third row); and complete (100%) remyelination (where the demyelinated segments from the third row were remyelinated by two shorter segments with 75% of lamellae). CVs averaged over 30 trials in each case. (C) Changes in CV (measured in %) in response to demyelination and remyelination versus the magnitude of current clamp step (+180, +280, or +380 pA). Shown are mean +/- SEM for demyelinating 50% of myelinated segments (removing all lamellae), and subsequent remyelination of those segments by shorter segments with 75% of lamellae.

The focus of most analysis is on the conduction velocity but in the end, this has no effect on WM, so the discussion of CV remains sterile.

CV delays likely do affect brain functions that rely on neuronal oscillations and synchrony, as mentioned in the Discussion. As such, we feel that our single neuron model results on CV delays as well as AP failures are valuable for the scientific community. Yet, given the results of our network models here, the reviewer has a valid point. We have clarified in the introduction that AP failures but not CV delays affected the network output (line 115):

“Higher degrees of demyelination led to slower propagation and eventual failure of APs along the axons of the multicompartment models. In the network models, an increase in AP failure rate resulted in progressive working memory impairment, whereas slower conduction velocities, in the range observed in the multicompartment models, had a negligible effect.”

We have also revised the single neuron section of the Results throughout, to better highlight the effects of myelin dystrophy on AP failures. Revisions to address this in the demyelination section start on line 148:

“AP propagation was progressively impaired as demyelination increased (Figure 3): CV became slower, eventually leading to AP failure. Removing 25% of lamellae had a negligible effect on CV, regardless of how many segments were affected. However, when all lamellae were removed, CV slowed drastically – by 38 ± 10% even when just 25% of the segments were demyelinated in this way, and 35 ± 13% of APs failed. When 75% of segments lost all their lamellae, CV slowed by 72 ± 8% and 45 ± 13% of APs failed.”

Similiarly, we have added several sentences about AP failures that remain after remyelination of the single neuron model (starting on line 190):

“Results for the percentage of AP failures (Figure 4C,F) were consistent with those for CV recovery. Remyelinating all previously demyelinated segments, even adding just 10% of lamellae, brought AP failure rates down to 14.6 ± 5.1%. Remyelinating all affected segments with 75% of lamellae (the maximal amount of remyelination) nearly eliminated AP failures (1.8 ± 1.1%). Incomplete remyelination, where some segments were still demyelinated, still had relatively high AP failure rates. For example, when one eighth of segments were remyelinated with the maximal amount of lamellae and one eighth were left bare, 25.7 ± 11.5% of APs failed across the cohort (Figure 4C, red dashed line and arrow). AP failure rates were slightly lower when starting with partial demyelination: 10.6 ± 7.6% of APs failed in the analogous paradigm (Figure 4F, red dashed line and arrow). In short: combinations of demyelinated and remyelinated segments often led to sizable CV delays and AP failures.”

The more important effect of de/re myelination is on failure. However, the failure is, AFAIK, just characterized by a constant current injection of 380pA. From Fig 2 it seems however that the first spike is particularly susceptible to failure. In other words, it has not been justified that it is fine to use the failure rates from this artificial protocol in the I&F model. I would expect the temporal current trace to affect whether the propagation fails or not.

In general, we did not find the first spike to be more susceptible to failure than latter spikes; the trace in Figure 2 is a representative snapshot intended to illustrate CV slowdown, AP failure, and recovery. Regarding the constant current injection: while the reviewer is correct that neurons do not receive such inputs in vivo, the applied current injections were designed to match in vitro current clamp protocols for these rhesus monkey neurons. While our future studies will include responses to more realistic synaptic inputs, we focused on somatic current injections here. We have added a new panel (C) to Supplementary Figure 1 (see previous response above) showing that the current step magnitude had little effect on the CV change after myelin perturbations; there was little effect on AP failure rates too. We now also state this finding more explicitly in Methods (starting on line 561):

“As done during in vitro electrophysiological experiments (Chang et al., 2005; Ibanez et al., 2020) and past modeling studies (Coskren et al., 2015; Rumbell et al., 2016), we first applied a holding current to stabilize the somatic membrane potential at -70 mV, then injected a current step into the somatic compartment for 2 seconds. …The CV changes in response to myelin alterations were relatively insensitive to variations in the magnitude of suprathreshold somatic current steps (Supplementary Figure 1C), and whether the current was constant or included Gaussian noise. Therefore, here we quantified CV changes and AP failures from responses to constant +380 pA current steps only.”

I don't know if there are many axon-collaterals in the WM circuits and or distance dependence in the connectivity, but if so, then the current implementation of failure would be questionable.

We agree that axon collaterals may affect our results; our unpublished morphological analyses of individual neuron axons indicate that there is a high degree of local axon collateralization in Layer 3 pyramidal neurons in LPFC. In this first study from our group on myelin perturbations, we chose to focus here on unbranched axons. There was some distance dependence of AP failure along the length of the axon. For example, in our most extreme demyelination case (75% of segments losing all their lamellae), about 14% of the axons showed more AP failure at their distal ends relative to the middle (mean difference 6.33%). We are examining this distance dependence more broadly in our next study, now cited in the Discussion (line 434): “Our model can also be extended to explore interactions between spatially localized myelin perturbations (such as those seen in multiple sclerosis) and axon collateralization (Sengupta et al., 2023), which would affect the distance-dependence of AP failures.”

I would also advise against thresholding at 75% failure in Fig3C. Why don't the authors not simply plot the failure rate?

We thank the reviewer for this suggestion, and have made this change. As suggested by the reviewer, we now show the AP failure rate in Figure 3 and Figure 4. The trends shown are nearly identical to those from the high failure trials.

Regarding the presentation, there are a number of dead-end results that are not used further on. The paper is rather extensive, and it would be clearer if written up in half the space. In addition, much information is really supplementary. The issue of the CV I already mentioned, also the Lasso regression for instance remains unused.

We understand the reviewer’s perspective, and we do value brevity when possible. During the revision process we examined the paper carefully, and made things more concise when it was feasible. As mentioned above, reporting CV results is important, though these revisions increased emphasis on results for AP failures in our revision. We combined the two Supplementary Figures about remyelination in the single neuron model into one (Supplementary Figure 3). We also moved the Lasso figure and associated methods to the Supplementary Material (Supplementary Figure 2), and have separated the Lasso results for demyelination and remyelination into their respective paragraphs (lines 154-160 and lines 200-204 respectively). While we do not use the Lasso explicitly later in Results, we cite them in the Discussion when comparing our findings to previous work (starting on line 417):

“Since our single neuron cohort sampled a wide range of parameter space, we used Lasso regression to identify which of the complex, interacting parameters contributed most to CV delays (which preceded AP failures). Parameters including axon diameter, node length, length of myelinated segments, and nodal ion channel densities predicted how our models responded to demyelination and remyelination; these findings are consistent with past modeling studies over more limited parameter ranges (e.g., Goldman and Albus, 1968; Moore et al., 1978; Babbs and Shi, 2013; Young et al., 2013; Schmidt and Knösche, 2019).”

We hope that our revision has struck an appropriate balance between clear and concise writing, and addressing concerns from both reviewers. We greatly value the time you have given to help us to improve our manuscript.

**Response to Recommendations for the Authors:**

**Reviewer #1 (Recommendations for the Authors):**
As I mentioned above, I consider that this study is well designed and it offers very interesting results. I have detailed below some of the issues that should be addressed to improve its potential impact in the field:(1) Across the manuscript, it is not entirely clear how the results of the multicompartmental model compare to existing modeling results on demyelination and CV changes (such as in the papers cited by the authors). Is this section confirming previous results with a new (more accurate) computational model, or are there any new insights previously unreported? A new paragraph in the Discussion putting these results in context would be very useful for the reader.

We thank the reviewer for this suggestion. We have added two new subheadings to organize the Discussion better, and have expanded the single neuron section to three paragraphs. We feel this now clarifies how our model fits in with previous work while stating its novelty more explicitly. Starting on line 391:

“Myelin changes affect AP propagation in a cohort of model neurons

The novelty of our neuron model lies in its systematic exploration of a combination of different myelin perturbation types known to occur in myelin dystrophies, across a wide range of biologically feasible models. Our single neuron model assumed that age-related myelin dystrophies (e.g., Figure 1) alter the insulative properties of lamellae analogously to demyelination, and examined interactions between demyelination and remyelination. Past studies of myelin dystrophy examined how either demyelination or remyelination of all segments affected AP propagation for a few representative axon morphologies. For example, Scurfield and Latimer (2018) explored how remyelination affected CV delays, finding that axons with more transitions between long and short myelinated segments had slower CV (Supplementary Figure 4), and was first to explore how remyelination interacts with tight junctions. However, their study did not couple remyelination and demyelination together or examine AP failures. Other basic findings from our single neuron cohort are consistent with past modeling studies, including that demyelination caused CV slowing and eventual AP failures (Stephanova
et
al., 2005; Stephanova
and
Daskalova,
2008; Naud
and
Longtin,
2019), and, separately, that remyelination with shorter and thinner myelinated segments led to CV slowing (Lasiene et al., 2008; Powers et al., 2012; Scurfield and Latimer, 2018). However, by assuming that some previously demyelinated segments were remyelinated while others were not, we found that models could have much higher AP failure rates than previously reported. Such a scenario, in which individual axons have some segments that are normal, some demyelinated, and some remyelinated, is likely to occur. We also found a few neurons in our cohort showing a CV *increase* after remyelination, which has not generally been reported before and is likely due to an interplay between ion channels in the new nodes and altered electrotonic lengths in the perturbed myelinated segments (e.g., Waxman, 1978; Naud and Longtin, 2019).

Since our single neuron cohort sampled a wide range of parameter space, we used Lasso regression to identify which of the complex, interacting parameters contributed most to CV delays (which preceded AP failures). Parameters including axon diameter, node length, length of myelinated segments, and nodal ion channel densities predicted how our models responded to demyelination and remyelination; these findings are consistent with past modeling studies over more limited parameter ranges (e.g., Goldman and Albus, 1968; Moore et al., 1978; Babbs
and
Shi,
2013;
Young
et
al.,
2013;
Schmidt
and
Knösche,
2019). Better empirical measurements of these parameters in monkey dlPFC, for example from 3-dimensional electron microscopy studies or single neuron axon studies combined with markers for myelin, would help predict the extent to which myelin dystrophy and remyelination along individual axons with aging affect AP propagation.

Another important feature of our multicompartment model is that it was constrained by morphologic and physiological data in rhesus monkey dlPFC —an extremely valuable dataset from an animal model with many similarities to humans (Upright and Baxter, 2021; Tarantal et al., 2022). While beyond the scope of the current study, this computational infrastructure –with a detailed axon, initial segment, soma, and apical and basal dendrites– enables simultaneous investigations of signal propagation through the dendritic arbor and axon. Our model can also be extended to explore interactions between spatially localized myelin perturbations (such as those seen in multiple sclerosis) and axon collateralization (Sengupta et al., 2023), which would affect the distance-dependence of AP failures. Integrating such results from single neuron models into network models of working memory, as we have done here, is a powerful way to connect empirical data across multiple scales.”

(2) Although the authors provide a well-designed study for the multi-compartmental model, it would be useful to add more details about how an unperturbed model and a completely remyelinated model differ in practice, perhaps right before the first results on the single cell model are presented. Are the new myelin sheaths covering the same % of axon as in the original case? Are there the same number of nodes? It is hard to distinguish which of these results are due to a compensation by the new myelin sheaths and which ones are just the model coming back to its original (and mathematically equivalent) starting point.

A ‘fully remyelinated’ axon is not mathematically equivalent to the unperturbed axon. Newly remyelinated segments had at most 75% of the original number of myelin wraps, with a new node in between, consistent with empirical observations in rhesus monkey dlPFC. Our manuscript changes in response to this recommendation are described in detail above in our response to the public review of the same reviewer.

(3) The authors observe a directed component in the bias that is known to be caused by heterogeneities in network connectivity, as stated in the text. It occurs to me that similar effects could be also caused by an heterogeneous demyelination in parts of the network. Inducing these biases could be another potential effect of demyelination in practice, and could be easily revealed by the author's current model (and displayed in a supplementary figure).

As suggested by the reviewer, we have tested heterogeneous demyelination in parts of the network and the results confirm the reviewer’s intuition. We have included these new results as new Supplementary Figure 7 (see below) and we have added the following sentences in the Legend of Figure 5, line 1265: “When demyelination is restricted to a part of the network, diffusion only increases in the perturbed zone (Supplementary Figure 7).” and in the Discussion (line 457): “In addition to age-related changes in memory duration and precision, our network model predicts an age-related increase in systematic errors (bias) due to an increased drift of the activity bump (Supplementary Figure 11). Moreover, if demyelination is spatially localized in a part of the network, the model predicts a repulsive bias away from the memories encoded in the affected zone (Supplementary Figure 7).”

**Author response image 5. sa3fig5:** Effect of spatially heterogeneous demyelination of the model neurons according to their preferred angle. We also tested working memory performance in the network when demyelination affects only parts of the network. The figure shows the decoded bump center position during the cue and delay period for the eight possible cue directions when a fraction of neurons was perturbed and the rest of the neurons in the circuit were unaltered (Figure 5B). We perturbed 10% of the neurons around the neuron with preferred direction 90° (left panel), 25% of the neurons around -90° (middle panel), and 50% of the neurons around 180° (right panel). Bump traces for cues that lie inside the perturbed portion of the circuit are shown in blue. Network perturbation in the three cases consisted in demyelinating 25% of the segments along the axons of model neurons, by removing 70% of the myelin lamellae. In each case, 280 trials were simulated for one network. These simulations show an increased drift and diffusion inside the perturbed zone, consistent with the increased drift and diffusion when perturbing the entire network (Figure 6B and Supplementary Figure 11). In particular, spatially heterogeneous demyelination in our network leads to a bias away from the affected zone and to increased trial-to-trial variability. Note that this is a model prediction, but we are not aware of empirical data showing heterogeneous demyelination with aging. Further, note that while our network model has a topological ring structure, neurons in PFC are not anatomically arranged depending on their preferred features. Thus, spatially heterogeneous demyelination would likely affect neurons with different feature preferences (i.e., neurons throughout our ring model).

(4) The bump attractor model of WM relies on a continuous attractor dynamics to encode the information stored in memory --a fixed point dynamics that can only vary via the slow noise-driven drift. This means, as the authors mention, that changes in CV won't affect the performance of WM in their model. This seems to be a limitation of the model, or at least an effect which is highly dependent on the modeler's choice, rather than an accurate prediction. While testing the effects of oscillations (as the authors argue in the Discussion) might be out of the scope of this work, there are other WM models which are more sensitive to temporal differences in activity. The authors should test whether the same (lack of) effects are also found in other WM models. A silent WM model seems to be the ideal candidate for this, as the authors already have the key dynamics of that model incorporated in their computational framework (namely, short-term synaptic facilitation in excitatory synapses).

We fully agree that considering the effects of demyelination in networks with alternative mechanisms would strengthen our manuscript. As suggested by the reviewer, we have simulated demyelination effects (AP failures and changes in CV) in an activity silent working memory model. The results are described in detail above in our response to the public review of the same reviewer.

We also would like to mention that we have now also tested larger conduction delays in the bump attractor model, revealing additional working memory errors. This is shown in the revised version of Supplementary Figure 6 (see below). However, those delays are unrealistically large and thus the main effect in both the bump attractor and the activity-silent model is due to AP failures.

**Author response image 6. sa3fig6:** Effect of propagation delays on control and perturbed networks. (A) Memory strength (left panels) and diffusion (right panels) for the young, control networks with zero propagation delays (blue solid line), as in Figure 5, and with propagation delays from a uniform distribution with a range between 0 and 100 ms (yellow dashed line). (B) Memory strength and diffusion for perturbed networks when demyelinating 50% of the segments along the axons of model neurons, by removing 60% of the myelin lamellae without delays (red solid line), and with delays from a uniform distribution with a range between 0 and 40 ms (gray dashed line) and between 0 and 85 ms (black dash-dotted line). The measures of working memory performance were calculated by averaging across 20 networks and 280 trials for each network. Shaded areas indicate SEM for each case. For the young, control networks, there was no difference with and without propagation delays, even though the delays used in the network simulations were much larger than the delays quantified in the single neuron model (the longest delays found for the most extreme perturbation condition –demyelination of 75% of the segments by removing 100% of the myelin lamellae– were of 49.9 ms on average; A). Working memory performance was also unaffected in the perturbed network with AP failures for delays ranging between 0 and 40 ms, also larger than the ones quantified in the single neuron model (for the case of 50% of the segments demyelinated by removing 60% of the myelin lamellae, the average delay in the cohort was 4.6 ms and the maximum delay was 15.7 ms; B). However, including extremely long delays of up to 85 ms did further impair memory compared to the impairment level introduced by AP failures alone (B).

(5) Impact of demyelination and remyelination on working memory: Could the authors explain here how these biologically detailed alterations are implemented in the bump attractor model? Is the CV and AP failure rate adjusted to the values produced by the multicompartment neuron model with these myelin alterations?

Yes, the reviewer is right, the CV and AP failure rate have been adjusted to the values produced by the multicompartment neuron model. To clarify this in the manuscript, we have restated the text as follows:

Lines 243 - 249 (Results):

To investigate how myelin alterations affect working memory maintenance, we explored in the network model the same demyelination and remyelination conditions as we did in the single neuron model. Because our network model consists of point neurons (i.e., without detailed axons), we incorporated CV slowing as an effective increase in synaptic transmission delays (see Methods). To simulate AP failures, we adjusted the AP failure rate to the values given by the single neuron model, by creating a probabilistic model of spike transmission from the excitatory presynaptic neurons to both the excitatory and inhibitory postsynaptic neurons (see Methods).

Lines 722 - 747 (Methods):

Modeling action potential propagation failures in the network. The network model is composed of point neurons without an explicit model of the axon. To effectively model the action potential failures at the distal end of the axons quantified with the single neuron model under the different demyelination and remyelination conditions, the AP failure rate was adjusted to the values produced by the single neuron model. To do this, we perturbed the 10 control networks by designing a probabilistic model of spike transmission from the excitatory presynaptic neurons to both the excitatory and inhibitory postsynaptic neurons. From the single neuron model, for each demyelination/remyelination condition, we quantified the probability of AP failure for each of the neurons in the control cohort, as well as the percentage of those neurons that shared the same probabilities of failure. That is, the percentage of neurons that had probability of failure = 0, probability of failure = 1 or any other probability. Then, we computed the probability of transmission, , and we specified for the corresponding percentages of excitatory neurons in the networks. Thus, in the network model, we took into account the heterogeneity observed in the single neuron model under each demyelination/remyelination condition.

Modeling conduction velocity slowing in the network. To explore the effect of CV slowing along the axons of model neurons, we simulated 20 young, control networks and 20 perturbed networks with AP failure rates adjusted for the case of single model neurons with 50% of the segments demyelinated along the axons by removing 60% of the myelin lamellae (we ran 280 trials for each network). Then, we added random delays uniformly distributed with a minimum value of 0 ms in both cases, a maximum value of 100 ms in the control networks, and a maximum values of 40 ms and 85 ms in the perturbed networks, in both the AMPA and NMDA excitatory connections to both E and I neurons (Supplementary Figure 6). These large values were chosen because we wanted to illustrate the potential effect of CV slowing in our network and smaller, more realistic, values did not have any effect.

(6) "We also sought to reveal the effect on working memory performance of more biologically realistic network models with AP transmission probabilities matched to both axons with intact and with altered myelin sheaths, as likely occurs in the aging brain (Figure 1). Thus, we ran network model simulations combining AP failure probabilities corresponding to groups of neurons containing intact axons and axons presenting different degrees of demyelination." I fail to see the difference with respect to the results in previous sections. Is it that now we have subnetworks in which axons are intact and subnetworks with significant AP failures, while before there was no topological separation between both cases? Please clarify.

In Figures 5 and 6 the AP failure rate of the neural population in the network simulations was matched to the AP failure rate of the cohort of single model neurons for each demyelination/remyelination condition. Since not all model neurons have equal features, a given condition produces different levels of impairment in its neuron. Thus, we quantified the probability of AP failure for each neuron in the control cohort, as well as the percentage of those neurons that shared the same probabilities of failure. Then, we computed the probability of AP transmission for the corresponding percentages of excitatory neurons in the networks. Thus, in the network model, we took into account the heterogeneity observed in the single neuron model under each demyelination/remyelination condition.

However, In Figures 7 and 8, we consider additional heterogeneity due to a different degree of demylination/remyelination of different neurons. Here, excitatory neurons in the network model are not perturbed according to a single demyelination/remyelination condition. Instead, we allowed that different percentages of excitatory neurons had AP failure rates corresponding to different demyelination/remyelination conditions: some were unperturbed, while others had different degrees of demyelination (Figure 7) and different degrees of remyelination (Figure 8). We have modified the text for clarification in several places.

First, when we describe the impact of demyelination on working memory, we already mention that (line 271): “In each of the 10 networks, we set the AP failure rate of the excitatory neurons according to the distribution of failure probabilities of the neurons in the single neuron cohort for the given demyelination or remyelination condition. Thus, we took into account the heterogeneity of demyelination and remyelination effects from our single neuron cohort (Figure 3A; Supplementary Figure 3). Note that this heterogeneity originates from differences in axon properties, but probabilities of failure for all neurons in the network correspond to the same degree of demyelination (Figure 6). We will also consider networks that contain different combinations of axons with either intact or perturbed myelin (Figure 7 and Figure 8).”

Second, we have combined the text describing Figures 7 and 8 under a single section title, which reads “Simulated heterogenous myelin alterations match empirical data” (line 334) and start this section with (line 337): “Up to this point we have studied network models with AP failure probabilities corresponding to a single degree of myelin alterations (i.e., with all excitatory neurons in the network having AP failure rates matched to those of the single neuron cohort for one particular demyelination or remyelination condition). Next, we sought to reveal the effect on working memory performance of more biologically realistic network models, where excitatory neurons in the networks were perturbed according to a combination of different demyelination or remyelination conditions. That is, we simulated networks with excitatory neurons having AP failure probabilities matched to both neuronal axons with intact and with altered myelin sheaths in different degrees, as likely occurs in the aging brain (Figure 1).”

(7) "Unexpectedly, our model indicates that compared to the performance of networks composed of neurons possessing axons with intact myelin sheaths, both demyelination and remyelination leads to an impaired performance." This conclusion is quite interesting, but I lack intuition from the paper as of why it is happening. In fact, the authors say in the Discussion that "complete remyelination of all the previously demyelinated segments with sufficient myelin, with fewer transitions between long and short segments, recovered working memory function." Would we then see a minimum and then an increase in memory duration in Figure 9B if we extended the X-axis until we hit 100% of new myelin sheaths?

This is a very important question that we have carefully addressed in Results and Discussion. We distinguish between two remyelination cases in the models. Complete remyelination: when all (100%) the previously demyelinated segments have been subsequently remyelinated, and incomplete remyelination: when less than 100% (25%, 50% or 75%) of the demyelinated segments have been remyelinated. Figure 6 (middle and right columns) shows the two cases (black lines for any percentage of lamellae added vs. colored lines): for 100% of the segments remyelinated, the network performance is nearly or completely (when enough lamellae are added) recovered to the young network performance. In fact, with the single neuron model we observe that (lines 192 - 193 in Results): “Remyelinating all affected segments with 75% of lamellae (the maximal amount of remyelination) nearly eliminated AP failures (1.8 ± 1.1%)”. However, incomplete remyelination recovers the performance compared to demyelination (middle and right columns in Figure 6 vs left column), but this performance is worse than the performance of the young networks. The single neuron model shows that (lines 194 - 197 in Results): “Incomplete remyelination, where some segments were still demyelinated, still had relatively high AP failure rates. For example, when one eighth of segments were remyelinated with the maximal amount of lamellae and one eighth were left bare, 25.7 ± 11.5% of APs failed across the cohort (Figure 4C, red dashed line and arrow).”

In Figure 9B (now Figure 8B), we combine intact axons with axons that are only partially remyelinated (i.e., incomplete remyelination). Extending the X-axis in Figure 8B until 100% of new myelin sheaths would not imply a minimum and a subsequent increase, but a continuous impairment: the more axons we perturb (remyelinate) the higher is the impairment compared to the young cases where all the axons are intact.

The sentence "Unexpectedly, our model indicates that compared to the performance of networks composed of neurons possessing axons with intact myelin sheaths, both demyelination and remyelination leads to an impaired performance.", now reads as (lines 379 380 in Results): “Therefore, both demyelination and incomplete remyelination lead to impaired performance in our networks, compared to networks with intact myelin sheaths”. We have also rewritten the corresponding section in Discussion (lines 486 - 489) as follows: “Therefore, it is reasonable to assume that ineffective remyelination may lead to working memory impairment. In fact, complete remyelination of all previously demyelinated segments with sufficient myelin, with fewer transitions between long and short segments, led to full recovery of working memory function.”

(8) [minor] "Our recent network model found that age-related changes in firing rates and synapse numbers in individual neurons can lead to working memory impairment (Ibañez et al., 2020), but did not consider myelin dystrophy." Could you be more precise about which age-related changes were studied in Ibanez et al. 2020? From the paper it seems like it was mostly cellular excitability and synaptic density, so this should be added here for more context.

To clarify this, we have added the following sentences in the Introduccion (line 105):

“Our recent network model revealed that the empirically observed age-related increase in AP firing rates in prefrontal pyramidal neurons (modeled through an increased slope of the *f*-*I* curve) and loss of up to 30% of both excitatory and inhibitory synapses (modeled as a decrease in connectivity strength) can lead to working memory impairment (Ibañez et al., 2020), but this model did not incorporate the known changes to myelin structure that occur during normal

aging.”

(9) [minor] "Recurrent excitatory synapses are facilitating, which promotes robust and reliable persistent activity despite spatial heterogeneities in the connectivity or in the intrinsic properties of the neurons." It would be great to add a reference here to justify the inclusion of this type of plasticity in the excitatory circuit (for example Wang, Markram et al. Nat Neuro 2006).

We have added the references suggested by the reviewer and a further one in the Results (line 216):

“Recurrent excitatory synapses are facilitating, as has been empirically observed in PFC (Hempel et al., 2000; Wang et al., 2006), which promotes robust and reliable persistent activity despite spatial heterogeneities in the connectivity or in the intrinsic properties of the neurons.”

References:

Hempel, C. M., Hartman, K. H., Wang, X. J., Turrigiano, G. G., and Nelson, S. B. (2000). Multiple forms of short-term plasticity at excitatory synapses in rat medial prefrontal cortex. J. Neurophysiol. 83, 3031–3041. doi: 10.1152/jn.2000.83.5.3031

Wang, Y., Markram, H., Goodman, P. H., Berger, T. K., Ma, J., and Goldman- Rakic, P. S.(2006). Heterogeneity in the pyramidal network of the medial prefrontal cortex. *Nat.Neurosci*. 9, 534–542. doi: 10.1038/nn1670